# Investigations into the Development of a Satellite-Based Aerosol Climate Data Record using ATSR-2, AATSR and AVHRR data over North-Eastern China from 1987 to 2012

Yahui Che[1,7], Jie Guang[1], Gerrit de Leeuw[2,3], Yong Xue[1,4], Ling Sun[5], and Huizheng Che[6]

[1] Key Laboratory of Digital Earth Science, Institute of Remote Sensing and Digital Earth, Chinese Academy of Sciences (RADI/CAS), Beijing 100094, China
[2] Finnish Meteorological Institute, Climate Research Programme, P.O. Box 503, FI-00101 Helsinki, Finland
[3] School of Atmospheric Physics, Nanjing University of Information Science and Technology, Nanjing 210044, China
[4] Department of Electronics, Computing and Mathematics, College of Engineering and Technology, University of Derby,
Derby DE22 1GB, UK
[5] Key Laboratory of Radiometric Calibration and Validation Environmental Satellites (LRCVES/CMA), National Satellite Meteorological Center, China Meteorological Administration, Beijing 100081, China
[6] State Key Laboratory of Severe Weather and Institute of Atmospheric Composition, Chinese Academy of Meteorological Sciences, CMA, Beijing 100081, China
[7] University of Chinese Academy of Sciences, Beijing, 100049, China
*Correspondence to*: Jie Guang (guangjie@radi.ac.cn) and Gerrit de Leeuw (Gerrit.Leeuw@fmi.fi)

**Abstract.** Satellites provide information on the temporal and spatial distributions of aerosols on regional and global scales. With the same method applied to a single sensor all over the world, a consistent data set is to be expected. However, the

application of different retrieval algorithms to the same sensor, and the use of a series of different sensors may lead to substantial differences and no single sensor or algorithm is better than any other everywhere and at any time. For the production of long-term climate data records, the use of multiple sensors cannot be avoided. The Along Track Scanning Radiometer (ATSR-2) and the advanced ATSR (AATSR) Aerosol Optical Depth (AOD) data sets have been used to provide a global AOD data record over land and ocean of 17-years (1995-2012), which is planned to be extended with AOD retrieved from a similar

sensor. To investigate the possibility to extend the ATSR data record to earlier years, the use of an AOD data set from the Advanced Very High Resolution Radiometer (AVHRR) is investigated. AOD data sets used in this study were retrieved from the ATSR sensors using the ATSR Dual View algorithm ADV v2.31 developed by Finnish Meteorological Institute (FMI), and from the AVHRR sensors using the ADL algorithm developed by RADI/CAS. Together these data sets cover a multi-decadal period (1987-2012). The study area includes two contrasting areas, both as regards aerosol content and composition

and surface properties, i.e. a region over North-East (NE) China encompassing a highly populated urban/industrialized area (Beijing-Tianjin-Hebei) and a sparsely populated mountainous area.

Ground-based AOD observations available from ground-based sun photometer AOD data in AERONET and CARSNET are used as reference, together with broadband extinction methods (BEM) data at Beijing to cover the time before sun photometer observations became available in the early 2000's. In addition, MODIS-Terra C6.1 AOD data are used as reference data set

over the wide area where no ground-based data are available. All satellite data over the study area were validated versus the reference data, showing the qualification of MODIS for comparison with ATSR and AVHRR. The comparison with MODIS

shows that AVHRR performs better that ATSR in the north of the study area (40°N), whereas further south ATSR provides better results. The validation versus sun photometer AOD shows that both AVHRR and ATSR underestimate the AOD, with ATSR failing to provide reliable results in the winter time. This is likely due to the highly reflecting surface in the dry season, when AVHRR-retrieved AOD traces both MODIS and reference AOD data well. However, AVHRR does not provide AOD larger than about 0.6 and hence is not reliable when high AOD values have been observed over the last decade. In these cases, ATSR performs much better, for AOD up to about 1.3. AVHRR-retrieved AOD compares favourably with radiance-derived AOD, except for AOD higher than about 0.6. These comparisons lead to the conclusion that AVHRR and ATSR AOD data records each have their strengths and weaknesses which need to be accounted for when combining them in a single multi-decadal climate data record.

## 1 Introduction

Aerosol particles are important atmospheric constituents, which play significant roles in many processes such as atmospheric chemistry, the absorption and scattering of solar radiation, and the lifetime of cloud and precipitation systems (Boucher et al, 2013; Koren et al., 2014; Guo et al., 2014; 2016a; 2018). Aerosol particles have an adverse effect on human health and are responsible for 7 million premature deaths annually over the whole world (WHO, 2018). Processes involving aerosols and their effects depend on the chemical and physical properties of the aerosol particles which in turn are determined by sources of directly emitted particles, the formation of aerosols from precursor gases (and thus the sources of these gases), the transformation of these particles during chemical and physical processes in the atmosphere and their removal by wet or dry deposition (cf. Seinfeld and Pandis, 1998, for a comprehensive treatment of aerosol processes). Observations of the concentrations of trace gases and aerosols are publicly available since several observational networks have been established, such as NASA's AERONET (AErosol RObotic NETwork) (Holben et al. 1998) (with observations mainly in the east of China), CARE-China (Xin et al., 2015), the Chinese Aerosol Remote Sensing Network (CARSNET) (Che et al., 2009; 2015) and SONET (Sun-sky radiometer Observation NETwork) (Li et al., 2018).  However, most of these observations started in the last decade and very few, if any, historical data on large scale are available for the construction of the long time series needed to show the evolution of pollutant concentrations over many years and analyse the effects of different contributions. Here, satellite data may offer a solution. The most common satellites used for the observation of trace gases and aerosols offer information since the beginning of the 21st century and, by combining the information from different instruments, time series encompassing two decades can be constructed (de Leeuw et al., 2018, Sogacheva et al., 2018b). Satellite information has been used together with model simulation to analyse the effects of natural and anthropogenic contributions on the concentrations of trace gases and aerosols (Kang et al., 2018). In another study combining satellite data with ground-based observations, the role of precursor gases, in particular VOCs, and photochemical reactions in the formation of aerosols ($PM_{2.5}$) was revealed (Bai et al., 2018).

In the second half of the 20th century the adverse effects of the precursor gases like $SO_2$ and $NO_2$ and aerosols on climate, air quality, and health were recognized and reduced by effective measures in developed countries. These led to the reduction of air pollution in developed countries, in particular in North America and Europe (Guerreiro et al., 2014), but in developing countries with increasing industrial activity and urbanization the concentrations continued to increase (Hao et al., 2000). As

an example, in China the concentrations of pollutants have increased over the years and are amongst the highest in the world. Recent publications show the effect of policy measures on the reduction of the most polluting trace gases $SO_2$ and $NO_2$ (van der A et al., 2017), which, as precursor gases, also affect the concentrations of aerosols. In particular, the emissions of $SO_2$ were reduced as part of the 11th Five-Year Plan (2006-2010) (Zheng et al., 2018), but the emissions of $NO_2$ continued to increase (e.g., van der A et al., 2017) until the 12th Five-Year Plan (2011-2015). Large emission reductions were achieved

after 2013 when the Clean Air Action was enacted and implemented and the $NO_2$ concentrations decreased (Zheng et al., 2018). Starting from 2011, aerosol concentrations decreased in China as shown, e.g., from satellite observations of the aerosol optical depth (AOD) (Zhang et al., 2017, Zhao et al., 2017, de Leeuw et al, 2018, Sogacheva et al., 2018b). However, early policies before 2005 on reducing pollution emissions are not effective, as shown from sparse ground-based observations (Jin et al., 2016). Long term satellite retrievals may offer a solution on a large spatial scale to observe how air pollutions control policies work and air pollution changes.

The Along Track Scanning Radiometer (ATSR-2), a dual view instrument, was launched in 1995 on the European Space Agency (ESA) satellite ERS-2 and provided data until 2003, and was one of the earliest satellite instruments used to retrieve Aerosol Optical Depth (AOD) quantitatively (Flowerdew and haigh, 1996; Veefkind et al., 1998). Its successor, the Advanced ATSR (AATSR) is a similar instrument launched in 2002 on the ESA platform ENVISAT, which was lost in April 2012. The

AOD over China from ATSR-2 and AATSR is consistent (Sogacheva et al., 2018a) and hence, together these instruments provide a 17-year AOD time series, 1995-2012 (Popp et al., 2016, de Leeuw et al., 2018). Combining ATSR and MODIS, 22-year AOD measurements were constructed, showing how the AOD increased until about 2006, and then clearly decreased since 2011 over China (de Leeuw et al., 2018, Sogacheva et al., 2018b). However, this time series should be extended especially before the 1990s when great changes have taken place in aerosols in China with the increase of industrialization. The Advanced

Very High Resolution Radiometer (AVHRR) onboard the NOAA satellites series could be a good choice to extend the ATSR time series as it started observations continuously from 1978 to the present. Xue et al. (2017) developed an AOD data set encompassinges two relatively small areas over Europe and China from 1983 to 2014, but covers the complete period, as opposed to the AVHRR global over land AOD data set recently presented by Hsu et al. (2017) and Sayer et al. (2017) which encompasses several distinct time periods. Hence in this study we focus here on the Xue et al (2017) AVHRR AOD data set

over China and compare that with ATSR-derived AOD data to determine its suitability for merging (as done for ATSR/MODIS by Sogacheva et al. 2018b), and thus extending the ATSR data set before the ATSR-2 era (and possibly after AATSR was lost in 2012, although other data sets may be more suitable for to extend to later years as shown in Sogacheva et al. (2018b) for MODIS. So the focus of the current study is to investigate whether the ATSR AOD data set can be extended to earlier years by using the Xue et al. (2017) AVHRR AOD data over China. For comparison of the ATSR and AVHRR AOD data sets, we

use both ground-based reference data, from AERONET (Holben et al., 1988) and from CARSNET (Che et al, 2009, 2015), and MODIS C6.1 AOD data. These reference data are not available for the period before 2000 and therefore we also use AOD data derived from broadband radiation measurements using the broadband extinction method (BEM) (Xu et al., 2015, Guo et al, 2016b) as described in Sect. 2.3.3. Data sets and methods used are presented in section 2. An overview of the data and an evaluation of their quality are presented in section 3, including a comparison of the various data sets. The results are discussed and conclusions are presented in section 4.

## 2 Method

### 2.1 Study area

The study area is located over north-eastern China, i.e. between 110° and 120° E and 35° and 45 °N (Fig. 1), which is divided into two sub-regions by the Taihang Mountain range with the North China Plain (NCP) and large urban agglomerations like Beijing and Tianjin and the Hebei province (together BTH which is among the most populated and fast developing regions in China) to the SE and the mountainous terrain to the NW extending over the Loess Plateau in Shanxi Province and the Inner Mongolia plateau. The Taihang mountain range forms a natural barrier for the transport of air pollution resulting in the frequent accumulation of pollutants and the occurrence of haze over the BTH area and the NCP (e.g., Sundström et al., 2012; Wang et al., 2013). The satellite-derived AOD maps in Fig. 2 show that this line also roughly divides high AOD in the SW of the study area and low AOD in the NW. The background in Fig. 1 is a land cover map showing that the major land cover types in the study area are croplands in the SE and grassland to the NW, which are intersected by mixed forest and closed shrublands as shown in the inset in Fig. 1.

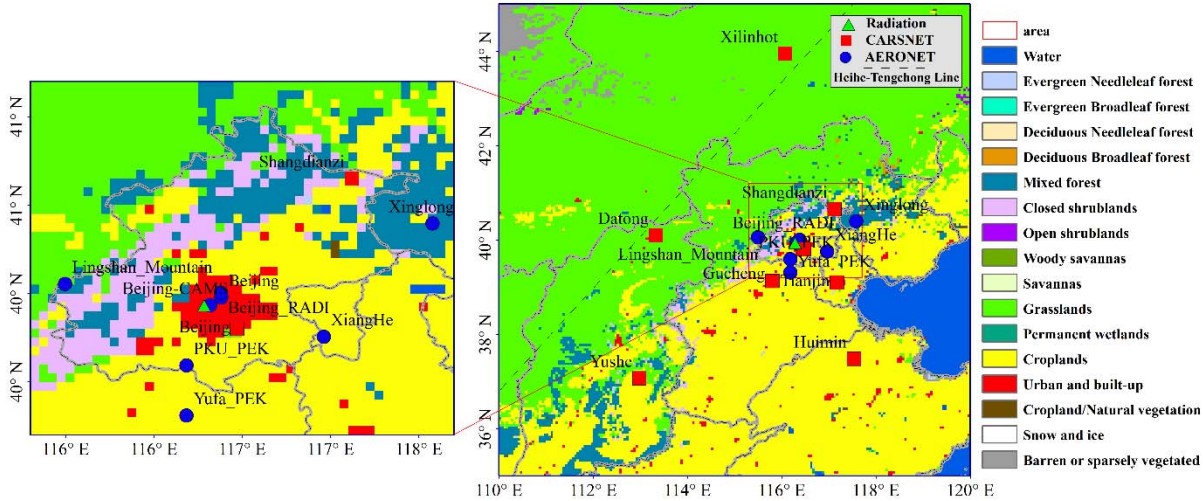

**Figure 1: Study area, with the locations of the ground-based reference sites discussed in Sect. 2.3 (CARSNET: red squares, AERONET: blue circles, solar radiation station: green triangle) overlaid on the IGBP land cover map.**

## 2.2 Satellite data

Satellite-retrieved data sets used in this study are satellite-derived AOD data from six radiometers, i.e. AVHRR-1, AVHRR-5 2, AVHRR-3, ATSR-2, AATSR and MODIS. These data sets are briefly discussed below.

### 2.2.1 AVHRR

The AVHRR instruments flew on a series of satellites, most of them with an afternoon equator crossing time at 13:40 LT (ascending) (see Xue et al., 2017, for an overview). AVHRR has a swath width of 2399 km at 833km altitude (Robel and Graumann, 2014) and thus provides daily global coverage. The AVHRR sensor was designed for measuring cloud cover and 10 surface temperature, but the observations are also used for the retrieval of AOD over ocean (e.g. Zhao et al., 2008), with only a few efforts to retrieve the AOD over land. Several AOD retrieval algorithms applied to AVHRR observations over land were published, including Hauser et al., (2005), Li et al., (2011), Mei et al., (2014), Xue et al. (2017), Sayer et al. (2017), Hsu et al (2017), and Gao et al., (2018). Xue et al. (2017) used the Algorithm for the retrieval of the aerosol optical Depth over Land (ADL) which was used to produce the continuous data set over China used in the current study. The use of AVHRR data for 15 aerosol retrieval requires a re-calibration of the radiances measured at the top of the atmosphere (TOA) because the sensors have no in-flight calibration. Xue et al. (2017) describe how this was achieved. Major problems in the retrieval of AOD from satellite observations is the effective decoupling of atmospheric and surface effects on the reflectance measured at the top of the atmosphere (TOA), cloud detection and the description of the aerosol properties. In the ADL algorithm a cloud mask is applied based on the Clouds from AVHRR (CLAVR, Stowe et al., 1991, 1999) and in the retrieval six aerosol types are used 20 as proposed by Govaerts (2010). For cloud-free pixels, and after application of gas absorption corrections as described in Xue et al. (2017), the land surface reflectance in the AVHRR channel 1 (0.64 μm) is parameterized in terms of the measured reflectance in Channel 3 (3.75 μm), with coefficients which are functions of the NDVI and scattering angle. The aerosol extinction at 3.75 μm is assumed to be negligible (which may not be true in the presence of coarse aerosol particles such as for desert dust). It is noted that the TOA reflectance at 3.75μm is composed of solar radiation reflected by the land/atmosphere 25 system and Earth radiation. The reflected part is estimated by using the method proposed by Allen et al. (1990). For each land cover type, as determined following the International Geosphere-Biosphere Programme (IGBP) land cover classification, a different parameterization of the 0.64 μm reflectance was developed. The IGBP land cover information used is the MODIS MCD12C1 product obtained from https://lpdaac.usgs.gov/dataset_discovery/modis/modis_products_table/mcd12c1, and is described in the MOD12 product ATBD (https://modis.gsfc.nasa.gov/data/atbd/atbd_mod12.pdf).

30 In the implementation of the ADL algorithm, the land surface reflectance is coupled to a radiative transfer model which includes individual parameterizations of the direct, single-scattered and multiple-scattered radiances. Thus, a model is developed for the TOA reflectance which is solved by optimal estimation. For more detail, see Xue et al., (2017). AOD is

retrieved at a wavelength of 0.64 μm at a spatial resolution of 0.05° × 0.05°. In this study AOD retrieved from NOAA-10 (1987-1991), NOAA-12 (1992-1998), NOAA-15 (1999-2002), NOAA-17 (2003-2009), and METOP-A (2010-2014) is used. The AVHRR data from 1983 to 1986 are not used because there are Jan to May in 1983, May to Jul in 1984, Jul to Oct in 1985, and Nov to Dec in 1986 are available, not consistent.

### 2.2.2 ATSR (ATSR-2 and AATSR)

Two Along-Track Scanning Radiometers (together referred to as ATSR) are used in this study: the ATSR-2, which flew on ESA's ERS-2 from 1995-2003, and the Advanced ATSR (AATSR), which flew on ESA's environmental satellite ENVISAT and provided data from May 2002 until April 2012. Both satellites flew in a sun-synchronous descending orbit with a day-time equator crossing time of 10:30 LT (ERS-2) and 10:00 LT (ENVISAT). Together these instruments provided 17 years of global aerosol data. The ATSR sensor has two views (near-nadir and 55° forward from nadir) which facilitate effective separation of the surface and atmospheric contributions to the reflectance at TOA. Multiple wavelengths (7) from VIS to TIR facilitate effective cloud screening and allow for multi-wavelength retrieval of aerosol properties. ATSR has a conical scan mechanism with a swath of 512 km, resulting in daily global coverage in 5- 6 days.

The ATSR dual view was first applied for AOD retrieval over land by Veefkind et al. (1998), based on the principles outlined by Flowerdew and Haigh (1995). Over ocean the two views are used separately to retrieve the AOD in both the nadir and forward directions. The ATSR dual view algorithm (ADV) has been much improved during algorithm experiments (Holzer-Popp et al., 2013) as part of the ESA Climate Change Initiative (cci) (Hollmann et al., 2013) project Aerosol_cci (de Leeuw et al., 2015, Popp et al., 2016). The most recent version of the ATSR dual view algorithm ADV is described in Kolmonen et al. (2016). ATSR L1 gridded brightness temperature data are provided with a nominal resolution of 1x1 km$^2$ sub-nadir and aerosol data are provided at a default spatial resolution of $10 \times 10$ km$^2$ on a sinusoidal grid (L2) and at 1° x 1° (L3).

The ATSR product used in this paper is the AOD at a wavelength of 550 nm, over the study area for the full ATSR mission. The data were produced using ADV version 2.31 which includes cloud post-processing as described in Sogacheva et al. (2017). ATSR-2 retrieved AOD data are available for the period June 1995- December 2003, with some gaps in 1995 and 1996. AATSR data are available for the period May 2002 - April 2012, but some data are missing in 2002 (see de Leeuw et al., 2018, for more details).

### 2.2.3 MODIS

The MODerate resolution Imaging Spectroradiometer (MODIS) flies aboard the NASA Terra and Aqua satellites, launched in December 1999 and May 2002, respectively, in a near-polar sun-synchronous circular orbit with an equator crossing time of 10:30 and 13:30 (LT), respectively (Salomonson et al., 1989). MODIS is a single view instrument with a swath of 2330 km (cross track) and provides near-global coverage on a daily basis. One of the most successful products of MODIS, which has been used in numerous aerosol related studies, is the AOD at 550 nm.

MODIS AOD is retrieved using two separate algorithms, Dark Target (DT) and Deep Blue (DB). In fact, two different DT algorithms are utilized, one for retrieval over land (vegetated and dark-soiled) surfaces (Kaufman et al., 1997, Remer et al., 2005, Levy et al., 2010, 2013) and one for retrieval over water surfaces (Tanré et al., 1997, Remer et al., 2005, Levy et al., 2013). The DB algorithm (Hsu et al., 2004, 2013) was traditionally used over bright surfaces where DT cannot be used (e.g. deserts, arid and semi-arid areas). However, the enhanced DB algorithm is capable of returning aerosol measurements over all land types (Sayer et al., 2013, 2014). The DT Expected Error (EE) is $\pm(0.05+0.15\tau_{AERONET})$ over land and $+(0.04+0.1\tau_{AERONET})$, $-(0.02+0.1\tau_{AERONET})$ over sea relative to the AERONET optical thickness ($\tau_{AERONET}$) (Levy et al., 2013). The DB expected error is $\sim\pm(0.03+0.2\tau_{MODIS})$ relative to the MODIS optical thickness ($\tau_{MODIS}$) (Hsu et al., 2013, Sayer et al., 2015). In this study the recently released (end of 2017) MODIS/Terra merged C6.1 L2 (10x10 km$^2$) AOD dataset is used because of the proximity of the MODIS/Terra and AVHRR overpasses. The merged (DT and DB) dataset is described by Levy et al. (2013) and includes measurements from both algorithms, offers a better spatial coverage and can be used in quantitative scientific applications (Sayer et al., 2014). This merged data set was validated over China using AERONET data, showing the good performance for all retrieved AOD values up to 2.4, with a small overall bias of 0.06 (Sogacheva et al., 2018a).

## 2.3 Ground-based reference data

Ground-based reference data used in this study are sun photometers from AERONET (Holben et al., 1998) and CARSNET (Che et al., 2015) available in the study area (see Fig. 1), complemented with Broadband Extinction Method (BEM) AOD data in Beijing for the period when no sun photometer data are available. The eight AERONET sites in the study area are all located in, or close to, Beijing. Three of them (Beijing, Beijing_RADI, and PKU_PEK) are located in the city, the others are located in rural areas in the vicinity of Beijing. The eight CARSNET sites are distributed over a wider area extending beyond BTH, with two of them (Xilinhot and Datong) located to the NW of the Heihe-Tengchong line, Huimin in the NCP and Yushe in rural areas surrounded by mountains. The other four (Lingshan_Mountain, XiangHe, Xinglong, and Yufa_PEK) are in and around Beijing. The solar radiation station is located in Beijing, close to three AERONET sites (Beijing, Beijing_RADI. Beijing_CMA).

The AERONET, CARSNET and radiation-derived AOD data sets are briefly described below.

### 2.3.1 AERONET

The AERONET project is a federation of ground-based remote sensing aerosol networks established by the National Aeronautics and Space Administration (NASA) in the USA and PHOtométrie pour le Traitement Opérationnel de Normalisation Satellitaire (PHOTONS) in France. AERONET was expanded with other networks and national efforts, see the AERONET website (https://aeronet.gsfc.nasa.gov/) for a description of contributors, sites, operational procedures, data products and availability, etc. AERONET serves as the primary network for global validation of satellite-retrieved aerosol products, including AOD which is used in this study. AOD at all AERONET stations is measured using CIMEL sun

photometers. Most common are the CIMEL CE-318 models with 5 wavelengths (440, 670, 870, 936 and 1020 nm) or polarized with 8 wavelengths, measuring direct sun as well at many angles using different scan patterns which provide the data necessary to retrieve a multitude of aerosol properties. Data checking and processing is done centrally and the products are freely available from the AERONET website. Data are made available at three levels, i.e. Level 1.0 (unscreened), Level 1.5 (cloud-

screened), and Level 2.0 (L2) (cloud-screened and quality-assured). The uncertainty of the CIMEL-derived AOD is 0.01-0.02 (wavelength dependent) (Eck et al., 1999). In this study we use AERONET L2 data from the most recent V3 dataset for validation and comparison (Giles et al., 2018). Satellite AOD data are commonly made available at 550 nm, and hence, for validation, the AERONET data are interpolated to this wavelength by using the Ångström Exponent (AE) which describes the AOD wavelength dependence (Ångström, 1924).

Almost all AERONET sites in the study area started observations after 2000 as Tab. 1 shows. Long-time and continuous measurements from 2001 to 2014 are only available from the Beijing and XiangHe stations. Therefore, we selected these stations for comparison with satellite time series, but for validation all available data are used.

**Table 1.** Data availability from AERONET sites in the study area (35°-45°N, 110°-120°E)

| | 2001 | 2002 | 2003 | 2004 | 2005 | 2006 | 2007 | 2008 | 2009 | 2010 | 2011 | 2012 | 2013 | 2014 |
|---|---|---|---|---|---|---|---|---|---|---|---|---|---|---|
| Beijing | √ | √ | √ | √ | √ | √ | √ | √ | √ | √ | √ | √ | √ | √ |
| Beijing-CAMS | | | | | | | | | | | | √ | √ | √ |
| Beijing_RADI | | | | | | | | | | √ | √ | | | |
| Lingshan_Mountain | | | | | | | | | | | | | | √ |
| XiangHe | √ | | | √ | √ | √ | √ | √ | √ | √ | √ | √ | √ | √ |
| Xinglong | | | | | | √ | √ | √ | √ | √ | √ | √ | | √ |
| PKU_PEK | | | | | | √ | | √ | | | | | | |
| Yufa_PEK | | | | | | √ | | | | | | | | |

### 2.3.2 CARSNET

AERONET sites in the study area are all located in or around Beijing (cf. Fig. 1). To expand the reference data set to other regions, AOD data from CARSNET (Che et al., 2015) was used, but only data for the years 2007, 2008 and 2010 were available for this study. The locations of the CARSNET sites in the study area are indicated in Fig. 1. CARSNET was established by the China Meteorological Administration (CMA) for the study of aerosol optical properties and validation of satellite retrievals, and the same instrumentation as AERONET (i.e. CIMEL CE-318) and similar procedures (Che et al., 2015). The difference is

the way of instrument calibration and AOD calculation as described in Che et al. (2009). The CARSNET AOD uncertainties are 0.03, 0.01, 0.01 and 0.01 at the 1020, 870, 670 and 440 nm channels (Che et al., 2009).

**2.3.3 Broadband extinction method (BEM) AOD data**

Broadband solar radiation has been measured in China by China Meteorological Administration (CMA) since the 1950s at 98 sites (Qiu, 1998) which evolved into the China national solar radiation network with 14 stations (Xu et al., 2015) providing continuous data since 1993 using China-made pyrheliometers. Methods were developed to retrieve monochromatic or equivalent AOD using hourly accumulated direct solar radiation and the results of these BEM AOD are in good agreement with sun photometer data (see Xu et al., 2015 for an overview). In the current study we used BEM AOD at 550 nm retrieved from the pyrheliometer measurements at the Beijing station (Fig. 1) of the China national solar radiation network. These data are particularly useful for evaluation of the AVHRR-retrieved AOD for the period before 2000 when no sun photometer data were available. The AOD was retrieved using BEM described by Xu et al. (2015). Xu et al. (2015) evaluate the quality of the BEM AOD (at 750 nm) for the Beijing station from comparison with AERONET AOD. The results show that 59% of the hourly AOD data fall within an error envelope of $\pm(0.05+0.15\tau_{AERONET})$. For monthly averaged data this is 82%. Guo et al (2016) used the hourly mean BEM AOD at 550 nm to evaluate the products from MODIS, OMI and MISR, obtaining consistent results with previous validations based on sun photometer measurements, which proved the effectiveness of the BEM AOD in satellite product validation.

**3 Results**

**3.1 Data overview**

The aim of the present study is to determine the compatibility of AVHRR and ATSR AOD data sets and their suitability to combine them to extend the ATSR time series to the 1980s. AVHRR AOD data is available over the study area from the ADL algorithm for the years 1983-2014 (Xue et al., 2017) and from ATSR-2 and AATSR using the ADV algorithm for the years 1995-2003 (ATSR-2) and 2002-2012 (AATSR), with some gaps as described in de Leeuw et al. (2018). The consistency between ATSR-2 and AATSR AOD is addressed in Sogacheva et al. (2018a) who show that no systematic differences occur over China. In the current work we also use MODIS/Terra C6.1 merged DTDB AOD for comparison. The validation of this data set over China by Sogacheva et al. (2018a), using all available AERONET stations, shows it's good quality, with a positive bias (0.06) with respect to AERONET AOD. As an example, AOD maps over the study area are shown in Fig. 2, seasonal aggregated for the full years 2000-2011 when data from all three sensors/algorithms are available: AATSR (ADV v2.31, L2, resolution 10x10 km$^2$), AVHRR (ADL, resolution 0.05º x 0.05 º) and MODIS C6.1 Merged DBDT (L2, resolution 10x10 km$^2$).

Figure 2 shows the similar AOD patterns derived from the three sensors/algorithms, with high AOD in the SE part of the study area and lower elsewhere. As indicated above, these high and low AOD areas are roughly separated by the Heihe-Tengchong line which separates the NCP from the Loess Plateau in Shanxi Province and the Inner Mongolia Plateau. In the SW of the study area, in Shanxi Province, we see an area with elevated AOD stretching from the NE to the SW, i.e. in the Guanzhong basin where pollution transported by NE winds from the BTH area accumulates between the Qinling Mountains and the Loess

Plateau and mixes with locally produced pollution. Comparison of the spatial distribution shows the similarity of the AOD distributions for AVHRR and MODIS while for AATSR the spatial distribution deviates with low AOD over Inner Mongolia, i.e. over the Gobi desert with high surface reflectance. ADV AOD retrieval is known to often be unsuccessful over bright areas and hence also aggregated AODs are too low (de Leeuw et al., 2018, Sogacheva et al., 2108a). AVHRR ADL appears to produce more credible AOD values in such conditions, as suggested by the similarity of the AOD patterns to those from MODIS. Below this statement will be put in perspective with the validation results from the ground-based reference data set.

Quantitatively, Figure 2 shows that in the south of the study area, i.e. south of 41°N, the MODIS AOD is overall higher than that retrieved from AATSR which in turn is overall somewhat higher than that from AVHRR. The smoother MODIS AOD, likely due to the larger number of data points because of the larger swath, together with the higher AOD and the scales chosen to plot the AOD maps, results in larger variability and patterns which are not as clearly revealed by ATSR and AVHRR. North from 41° AATSR AOD retrievals are often not successful so a quantitative comparison cannot be made at these latitudes. As regards the comparison of the other two sensors north of 41°N MODIS is overall higher than AVHRR, except, for instance, over an area in Inner Mongolia just north of 41°N where the MODIS AOD is close to zero while AVHRR provides AOD values between 0.1 and 0.2.

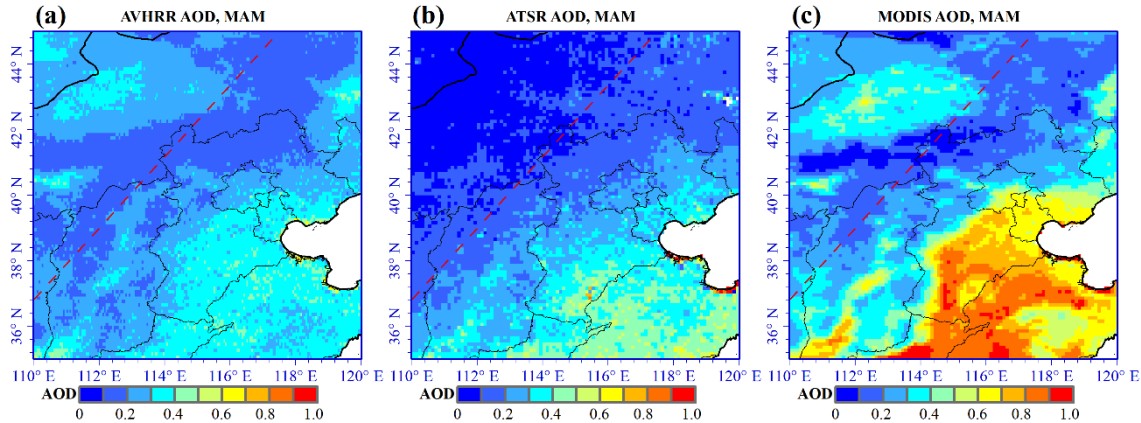

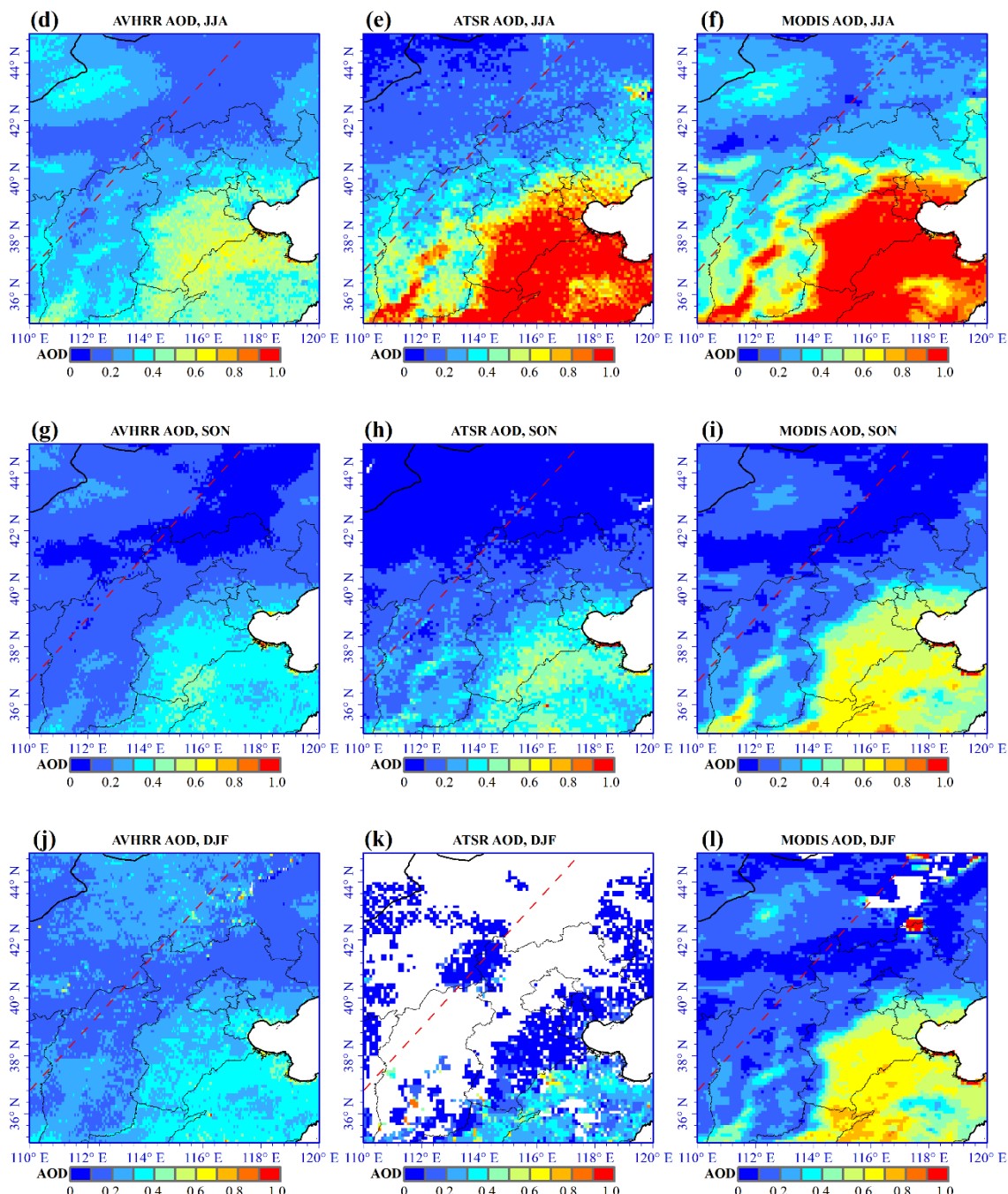

**Figure 2: AOD over the study area retrieved from AATSR (ADV v2.31, L2, 10x10 km2), AVHRR (ADL, 0.05 º x 0.05º) and MODIS C6.1 Merged DTDB (MODIS C6.1, L2, 10x10 km2), seasonal aggregated over the years 2000-2011.**

Below the satellite AOD data will be validated versus the ground-based reference data from AERONET and CARSNET, for all stations available in the study area, as indicated above. The AVHRR data will also be compared with the radiation-derived

AOD data available from the Beijing station which are especially useful for the earlier period when no other reference data are available. Next the satellite-derive AOD time series will be compared with the reference data to evaluate for each of the data sets when they are most useful. For AVHRR and MODIS, with wide swath, this will be done using monthly averaged AOD. For ATSR, with a much smaller swath, the data volume is too small for monthly averaging in a statistically meaningful sense
and therefore seasonal averages will be used. In the direct comparison of monthly time series from other sensors, this may lead to an apparent shift in the AOD peak values.

## 3.2 Data quality: validation

### 3.2.1 Procedure

The MODIS and ATSR AOD data sets used in this study have been validated versus sun photometer reference data, on global
and regional scales. With a focus on China, the ATSR v2.31 and MODIS C6.1 have been validated using AERONET data, for mainland China, for ten regions across China and for different seasons (de Leeuw et al., 2018, Sogacheva et al., 2018a). However, as shown in de Leeuw et al. (2018) and Sogacheva et al. (2018a), strong regional differences occur in both the seasonal and long-term AOD variations. Therefore, to achieve an unbiased comparison between the different data sets over the current study area, the MODIS C6.1 and ATSR data used in this study are validated versus the available reference data,
together with the AVHRR data. The ADL data over the study area was earlier validated by Xue et al. (2017), but only AERONET stations were used while in the current study the complementary information (AOD) from CARSNET is also used.

Collocation of satellite and reference data is important for validation. Here we follow the spatio-temporal collocation method proposed by Ichoku et al., (2002), i.e. the satellite data were averaged over an area of 5 x 5 pixels (ca. 50 x 50km$^2$ at nadir) around the sun photometer location, whereas the sun photometer data were averaged over ±30 min around satellite overpass
time. This spatio-temporal collocation method has been widely used for the validation of satellite aerosol products, for instance for MODIS (Ichoku et al., 2002, Chu et al., 2002, Remer et al., 2005, Levy et al., 2010, Sayer et al., 2014), AATSR (Che et al., 2016), and AVHRR (Riffler et al., 2010, Xue et al., 2017).

In the data presentation and discussion, expected error (EE) envelopes are used which apply to MODIS DT over land, i.e. expected errors of ±(0.05+0.15$\tau_{AERONET}$) (Levy et al., 2013). For comparison, this value is also used for ATSR and AVHRR
although the actual expected error envelopes for these sensors, which were not designed for aerosol retrieval, is expected to be higher than for MODIS. For ATSR a per-pixel uncertainty is provided rather than EE (Kolmonen et al., 2016) and for the AVHRR ADL (Xue et al., 2017) uncertainties have not been estimated.

The Beijing solar radiation station is located to the SW of the Beijing AERONET site at a distance of approximately 20km. AOD differences at these two sites are likely to occur due to emissions and secondary formation processes over this urban area,
but these have not been accounted for in the current study where collocations include averages over areas larger than the

separation between these sites. Therefore, BEM AOD is directly compared to the sun photometer AOD as was done by Xu et al. (2015).

**3.2.2 Results**

Figure 3 shows scatterplots of AOD, for each of the three data sets individually, versus ground-based AOD reference data
from all AERONET and CARSNET (3 years: 2007, 2008 and 2010) sites in the study area. The red symbols represent the averaged satellite-retrieved AOD binned in 0.1 sun photometer AOD intervals and the vertical lines on each circle represent the 1-sigma standard deviation of the fits. The colour bar on the right indicates the number of data points. Parameters are presented in the upper left corner, where N is the total number of collocated pairs and RMS is the root mean square error. The blue line represents the identity line and the black dotted lines represent the MODIS EE. Large differences occur between the
three data sets. Starting with MODIS/Terra C6.1 merged DTDB, which has the largest number of collocations, the scatterplot in Fig. 3a shows the excellent performance for AOD up to 1.3 with the bin-averaged AOD less than 0.05 below the identity line and 69.5% of all data points within expected error of $\pm(0.05+0.15\tau_{AERONET})$. For AOD of 1.3 and larger the bin-averaged AOD values are much lower than the reference values, although still within one standard deviation. For AOD>2.6, MODIS does provide values but they are all well below the identity line. These results indicate that MODIS C6.1 AOD over the study
area is reliable for AOD up to 1.3, but for higher AOD their use is not recommended.

ATSR with a swath width of 500 km provides much less collocations than MODIS but the large difference in performance (compare Fig. 3a and 3c) cannot be explained by the swath alone and is likely due to the failure of ADV to provide adequate retrievals over bright surfaces. Figure 3b shows that ADV provides AOD values up to 2.7 but for AOD > 1.3 the values are widely scattered around the identity line with large deviations from the sun photometer values. Also for lower AOD a
systematic underestimation is observed which increases with the AOD value up to 0.8 where the ADV-retrieved AOD is about 0.3 low. For larger AOD, up to 1.3, ADV continues to underestimate the AOD with a similar amount. Although 52% of the data points are within the EE, ADV clearly underestimates the AOD over this area and the bin-averaged AOD follows the lower EE limit rather than the identity line, for AOD up to 1.0, leading to the conclusion that ADV AOD is about 0.15 x AOD low.

AVHRR with a swath width of 2900 km is expected to provide the largest number of collocations but actually it is about 20% lower than for MODIS, likely due to failure of ADL to provide a valid retrieval in all situations. ADL does not provide retrievals for AERONET AOD > 2 and the data in Fig. 3a show the large scatter around the identity line and the large underestimation for AOD > 0.6. For AOD larger than about 0.8, the binned ADL AOD data deviate from the identity line by more than the expected error. For AOD up to 0.6, the bin-averaged values are within the EE (58.1% of the data), but
systematically deviate from the identity line and underestimation increases with increasing AOD.

In conclusion, in the study area, MODIS provides reliable AOD for values up to 1.3 with a slight underestimation. This is in contrast with the findings of Sogacheva et al. (2018a) who observed a slight overestimation (bias 0.06) of the MODIS/Terra

C6.1 merged DTDB AOD using all AERONET data available over China, but no CARSNET. CARSNET could provide more reference data in regions where AERONET data are not available, i.e. outside the Beijing region. For ATSR ADV v2.31 data the validation presented in de Leeuw et al. (2018), also using only AERONET data similar to Sogacheva et al. (2018a), deviates from that presented here, with a slight underestimation resulting in a bias of 0.07. Clearly the selection of the smaller study area affects the validation results, with for MODIS slightly better results (smaller bias) than over all China, but for ATSR ADV the performance is less good. AVHRR AOD shows a large scatter and does not follow the identity line even for low values, underestimates for AOD larger than 0.5 and appears to fail for AOD larger than about 0.8.

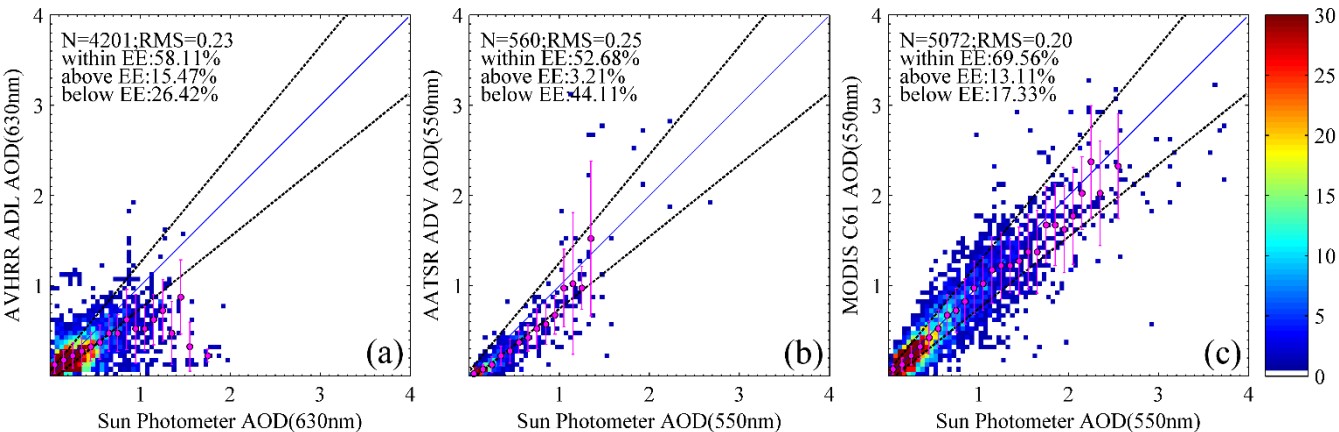

**Figure 3: Scatter density plots of satellite-retrieved AOD versus sun photometer retrieved AOD data from AERONET (all available data) and CARSNET (3 years: 2007, 2008 and 2010) stations in the study area.**

**Table2.** Metrics obtained from the evaluation of three AOD products. N is number of collocated data points, MSA is mean satellite retrieved AOD, MAA is mean sun photometer (AERONET and CARSNET) derived AOD, RMB is the ratio MAA/MSA. MBE is mean of AOD bias given by $\tau_{sat} - \tau_{aero}$. R is the Pearson correlation coefficient. RMS is the root mean square error. EE is the expected error, EE_a is the fraction above the EE, EE_b is the fraction below the EE.

| Product | Network | N | MSA | MAA | RMB | MBE | RMS | R | EE | EE_a | EE_b |
|---------|---------|------|-------|-------|-------|--------|-------|-------|--------|--------|--------|
| MODIS | AERO&CARS | 5072 | 0.441 | 0.464 | 0.950 | -0.023 | 0.204 | 0.910 | 69.56% | 13%.11 | 17.33% |
| ADV | AERO&CARS | 560 | 0.289 | 0.396 | 0.729 | -0.107 | 0.252 | 0.843 | 52.68% | 3.21% | 44.11% |
| AVHRR | AERO&CARS | 4201 | 0.237 | 0.296 | 0.802 | -0.059 | 0.229 | 0.602 | 58.11% | 15.47% | 26.42% |

Seasonal scatterplots are presented in the Appendix and discussed in the context of the seasonal variations observed in the AOD time series presented below.

**3.3 Comparison of satellite-derived AOD time series with reference data**

In this section time series of satellite-retrieved AOD data will be compared with time series available from the AERONET sites in Beijing and XiangHe, for which the longest time series are available, and with AOD derived from broadband radiances measured at the Beijing radiation station.

**3.3.1 AERONET AOD in Beijing**

AOD time series for the period 1997-2012, retrieved from AVHRR, ATSR and MODIS data and spatially collocated with the Beijing AERONET site, are presented in Fig. 4 together with monthly-averaged AERONET AOD for reference. The time series presented here start in 2000 because for the earlier years no AERONET and MODIS data are available. The data in Fig. 4 show that AVHRR is almost always lower than AERONET and that the high AOD values obtained from AERONET, usually in the summer, are not reproduced by AVHRR. This is not unexpected since the scatterplots in Fig. 3 show that AVHRR fails to retrieve high AOD. For these individual data points hardly any values larger than 1 were retrieved and hence the monthly averages in Fig. 4 are lower than that. Figure 4 shows that the highest AVHRR AOD values of about 0.8 occur in 2002 and 2003 and thereafter do not exceed 0.6. The monthly averaged AERONET AOD peak values occur mostly in the summer and are about 1.4. However, in other seasons the differences are much smaller and the two time series seem to trace rather well with an offset of about 0.1. In particular in the winter AVHRR AOD is close to the reference value. These seasonal differences are confirmed by the scatterplots presented in the Appendix. The results lead to the conclusion that, for the Beijing site, for low AOD the ADL algorithm provides quite reasonable results but for high AOD improvement is needed to provide reliable time series.

The MODIS data in Fig. 4 (top) trace the AERONET AOD very well, both in the summer and in other seasons, with MODIS AOD often just a little smaller as can also be observed from the scatterplots in the Appendix. This shows that MODIS serves as a good reference for the situation encountered at the Beijing site.

This does not apply to the ATSR AOD. Although ATSR AOD variations follow those from AERONET, and even reproduce the high summer values in some years (2007 and 2011), the ATSR values are most of the time smaller and especially in the winter the ATSR AOD is much too low. As discussed before, this may be due to the failure of ADV to produce a valid retrieval over bright surfaces.

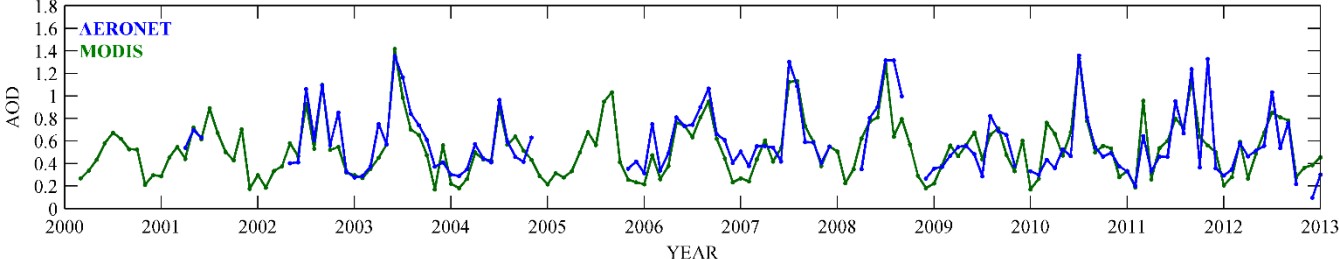

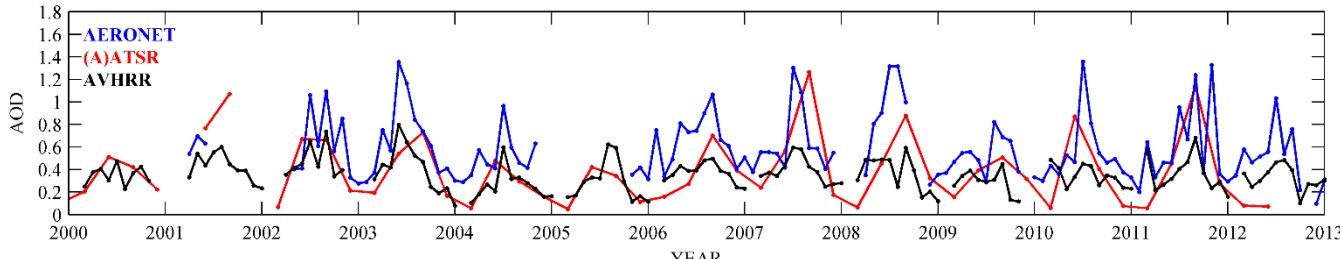

**Figure 4: AOD time series for MODIS (top), (A)ATSR and AVHRR (bottom), with AERONET AOD (blue) as reference, over the Beijing site for the years 2000-2012. For (A)ATSR seasonally averaged AOD is plotted, for AVHRR and MODIS the AOD data are monthly averages.**

### 3.3.2 AERONET AOD in XiangHe

Figure 5 shows a comparison of the AOD time series over the XiangHe AERONET site, similar to that over Beijing in Fig. 4. It is noted that, for comparison, in both cases the time series are plotted for the years 2000-2012, but that the XiangHe time series start at the end of 2004 (except for a few data points in 2001), i.e. later than in Beijing. Comparison of the AERONET time series shows that over XiangHe the AOD in the summer is somewhat higher and peaks often in the same years as over Beijing, but there are also differences such as in 2008 when the AOD in XiangHe reached the maximum monthly-averaged value of 1.5. As in Beijing, the AVHRR retrieval did not reproduce these high values in the summer, but for the other months the AVHRR/AERONET comparison is better than in Beijing and, with some exceptions, the AOD data trace very well. The seasonal scatter density plots in the Appendix, Fig. A3, confirm that no high AOD is retrieved in the summer, as opposed to other seasons. These scatterplots also confirm the underestimation of ADL, by about one EE, in each of the seasons except in the winter where ADL performance compares better to AERONET.

In contrast, the ATSR retrieval algorithm does reproduce the higher values, and in some years the seasonally averaged AOD compares favourably with the AERONET AOD (e.g. in 2007 and 2011), but in other years, such as 2008 and 2010 the maximum AOD is not observed by AATSR, which in part may be due to the seasonal averaging. Also, as over Beijing, ATSR retrieval does not work well in the winter time, possibly due to the high surface reflectance in the dry season, and the seasonal AOD is much lower than AERONET.

For MODIS the AOD compares well with AERONET but not as good as over Beijing. During some periods, e.g. in 2005 and during 2008-2011, MODIS is too high. This is also shown by the validation results in the Appendix. The reason for this difference between Beijing and XiangHe, which is located close to Beijing to the ESE (see Fig. 1), has not been further investigated. Possibly, the surface properties, urban and build-up for Beijing versus cropland for XiangHe, may affect the retrieval results as was also indicated in de Leeuw et al. (2018), but for MODIS C6.0.

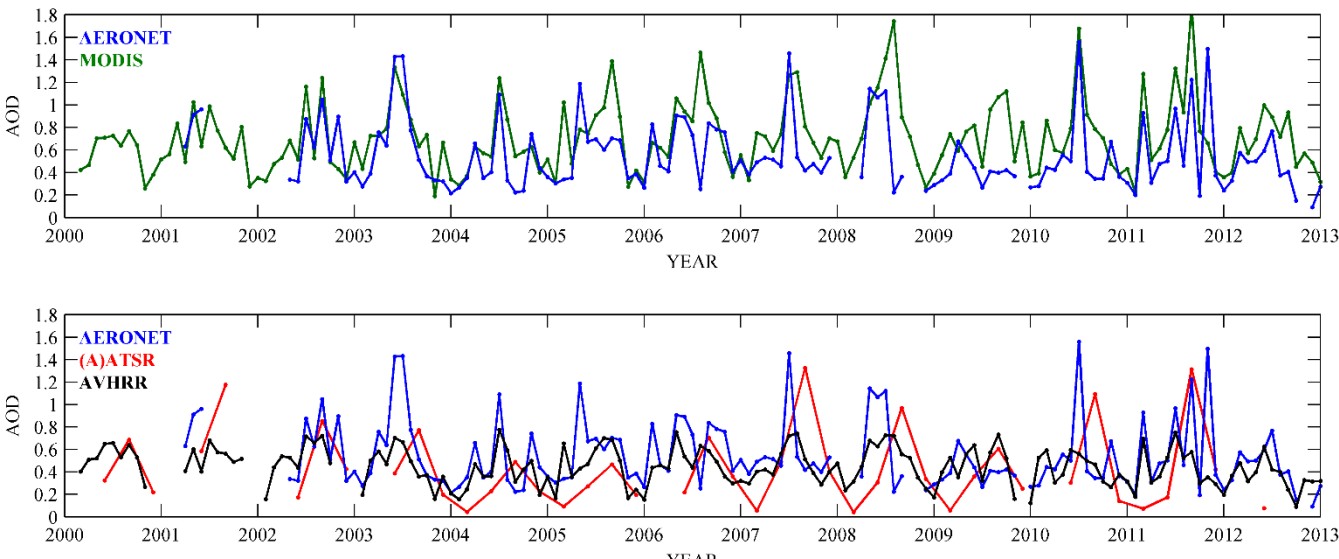

**Figure 5: AOD time series for MODIS (top), (A)ATSR and AVHRR (bottom), with AERONET AOD (blue) as reference, over the XiangHe site for the years 2000-2012. For ATSR seasonally averaged AOD is plotted, for AVHRR and MODIS the AOD data are monthly averages.**

### 3.3.3 Broadband extinction method (BEM) AOD in Beijing

Figure 6 shows the comparison of the monthly-averaged AOD time series from AVHRR with those derived from BEM data, as well as a comparison of the BEM AOD with AERONET and with MODIS, as reference data. Monthly radiance data at the Beijing site have been validated by Xu et al. (2015) for the years 2002-2012 using both MODIS C5 L3 AOD and AERONET L2 data (version 2). In the current study MODIS C6.1 and AERONET version 3 are used, for the years 2000-2012, and the results from the comparison of the BEM AOD with these data sets, shown in Fig. 7 (left), are similar to those presented in Fig. 2 of Xu et al. (2015). Also the time series in Fig. 6 show very good comparisons between the BEM and MODIS and AERONET AOD. Hence the BEM AOD data provide another reference data set for the AVHRR-retrieved AOD which can be used for the period 1987-2012 as presented in Fig. 8. Figure 8 shows that in this earlier period (1987-2000) the AOD was generally lower and the monthly averaged AOD compares better with BEM AOD than for the later period 2000-2012, when AOD was higher during especially the summer months. This is consistent with the data in the scatter plot in Fig. 7, right hand side, which shows that almost all of the scatters are below the identity line when AOD is larger than 0.6. The overall conclusion remains the same, however, that the AVHRR retrieval needs improvement for high AOD and only values up to about 0.6 can be used.

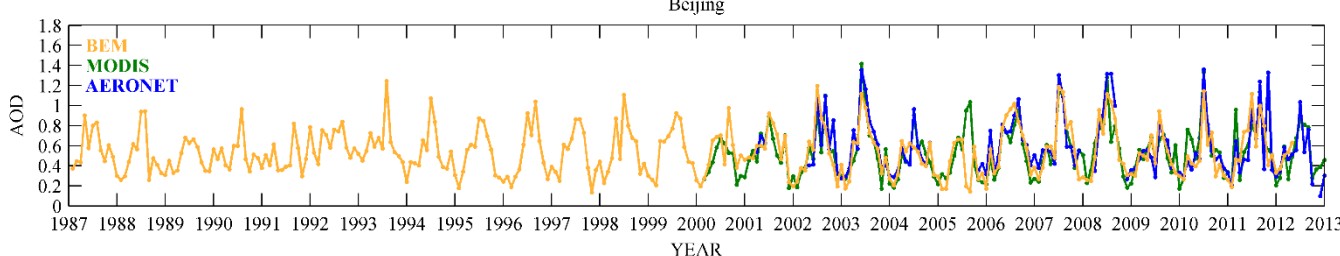

**Figure 6: AOD time series for BEM (orange), with MODIS (green) and AERONET AOD (blue) as reference, over the Beijing site for the years 1987-2012.**

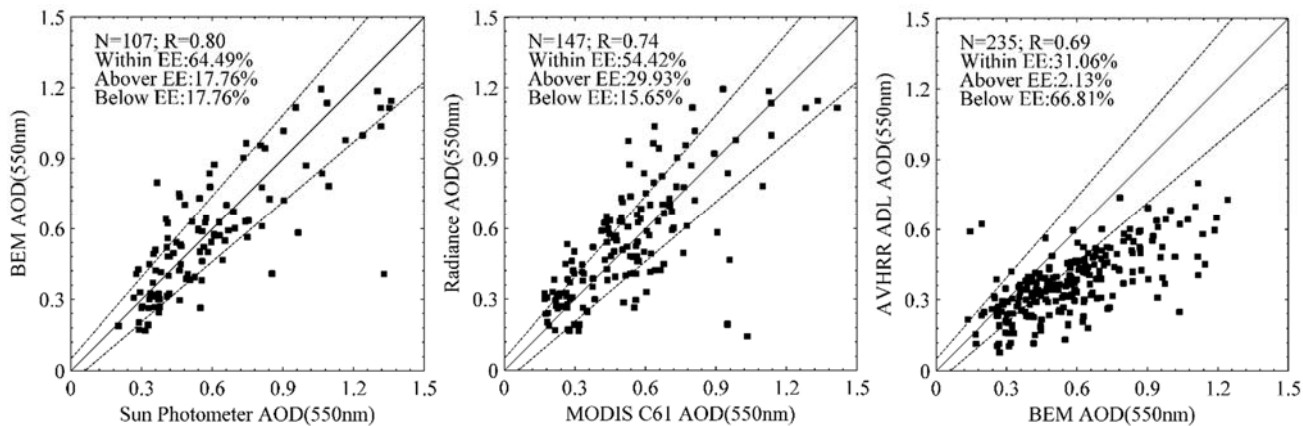

**Figure 7: Scatter plots of BEM AOD versus MODIS AOD, sun photometer AOD, and AVHRR AOD. The dashed lines show the MODIS expected error of ±(0.05+0.15τ_{AERONET}).**

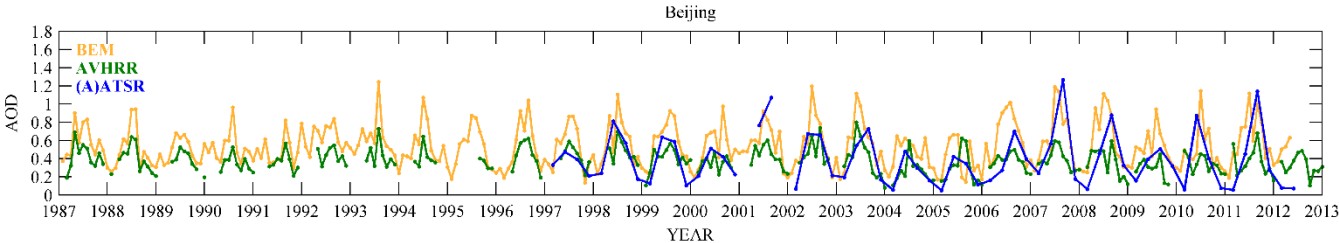

**Figure 8: AOD time series for BEM (orange), with AVHRR (green) and (A)ATSR AOD (blue), over the Beijing site for the years 1987-2012.**

### 3.4 Seasonal comparison of satellite-derived AOD

The above results and those from Sogacheva (2018a) and de Leeuw (2018) show the high quality of the MODIS data over China. Therefore, we select MODIS as reference data for comparison with both time series and spatial coverage. Time series

of satellite and ground-based AOD data were compared in the previous Section, here we compare the spatial distributions of AVHRR, ATSR and MODIS AOD.

As Fig. 2 shows, in the SE of the study area MODIS-retrieved AOD is often higher than 0.7 whereas the AOD retrieved from ATSR and AVHRR data are substantially lower due to failure of the algorithm to retrieve AOD larger than 0.6 (AVHRR) or unsuccessful retrieval and / or underestimation of the AOD by ADV except in the summer. Hence, the AOD difference patterns between the three instruments, presented in Fig 9b&c, show similar features in the SE of study area. However, in the spring the MODIS AOD is approximately 0.2 larger than that from ATSR and AVHRR. In the summer, when the MODIS AOD in the SE of the study area is very high with values larger than 0.8, the ATSR-retrieved AOD is similar to that from MODIS but the AVHRR-retrieved AOD is much lower (Fig. 9f). This indicates that AVHRR ADL algorithm needs to be improved for these conditions. In the autumn, the situation in the SE of study area is similar to that in the summer, i.e MODIS AOD is high with values exceeding 0.6 and ATSR AOD is close to that from MODIS with a difference of less than 0.1, while AVHRR ADL is lower due to failure to retrieve elevated AOD. In the winter season, ADV fails to provide successful retrievals over approximately half of the study area (mainly over bright surfaces) but over other areas the ATSR-retrieved AOD is close to that from AVHRR and MODIS as shown in Figs. 9j&k. It should be noticed that, in addition to algorithm failure over bright surfaces, the number of ATSR samples is much smaller than that of AVHRR and MODIS due to the much smaller swath width, and therefore the seasonally averaged AOD is less smooth than that from the other two sensors.

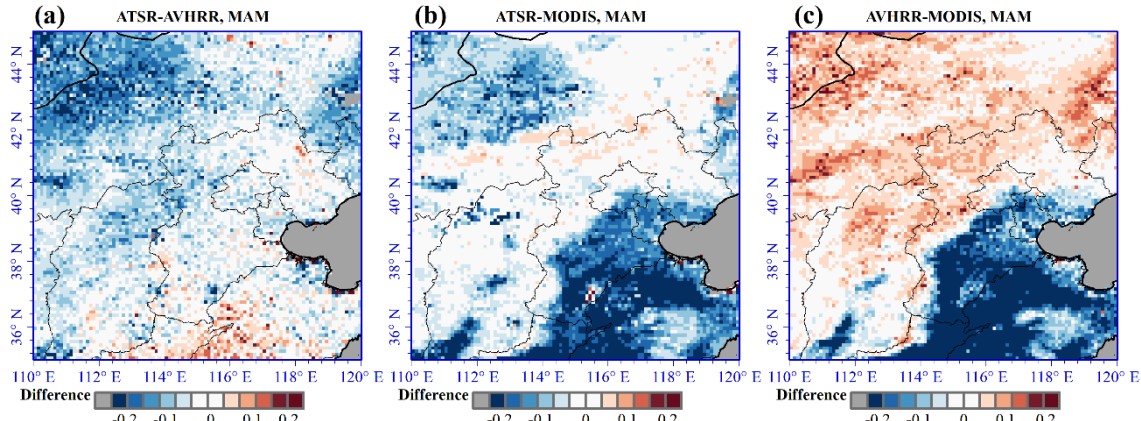

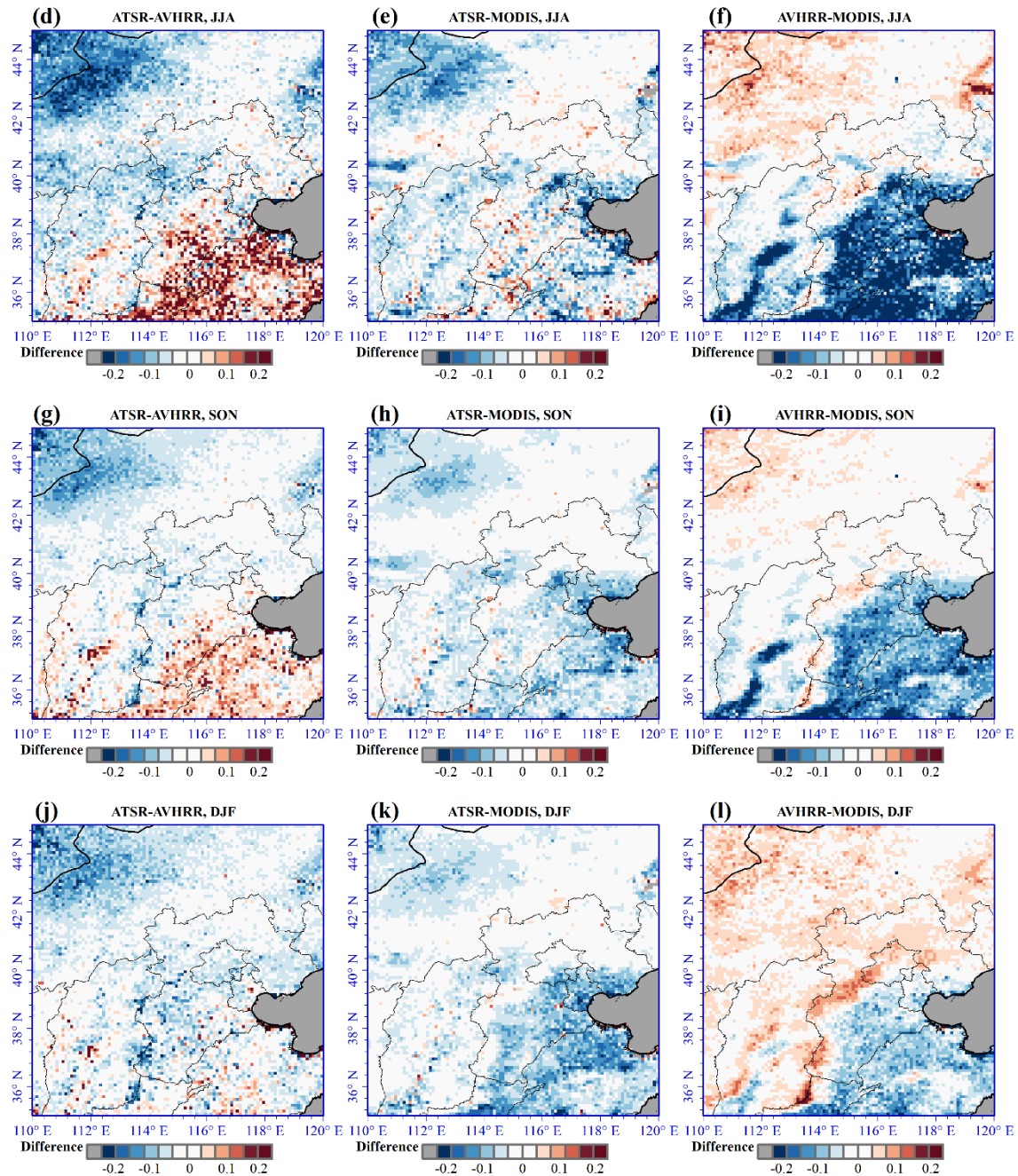

**Figure 9: AOD difference over the study area retrieved from AATSR (ADV v2.31, L2, 10x10 km²), AVHRR (ADL, 0.05 º x 0.05º) and MODIS C6.1 Merged DTDB (MODIS C6.1, L2, 10 x 10 km²), seasonal aggregated over the years 2000-**
10 **2011.**

## 4 Discussion and conclusions

The validation exercises and time series comparisons presented in Sect. 3 show the stronger and weaker points of the ATSR- and AVHRR-retrieved AOD Climate Data Records (CDRs), with AVHRR being better in the winter and ATSR better in the summer. These differences are also very clear in the direct comparison of the AVHRR and ATSR time series, shown in Fig.
5 10 for the Beijing site and in Fig. 11 for the XiangHe site. Figure 10 also shows that before 2007 these time series trace quite well and could be combined into a single time series while taking into account the validation results for that period only. In particular the use of the ATSR winter AOD data will have to be considered carefully and may possible have to be given little weight. However, after 2007 large differences are observed with quite high ATSR AOD in the summer while that retrieved from AVHRR has a maximum value of about 0.6. The comparison with MODIS shows that the ATSR AOD values are credible
10 and follow those of MODIS (apart from the shift due the use of seasonal ATSR vs monthly MODIS AOD averages), whereas the AVHRR AOD is much too low. In the winter the opposite is observed: ATSR is too low with values close to zero while AVHRR traces MODIS very well. The intercomparison of the ATSR and AVHRR time series for XiangHe in Fig. 11 shows a similar pattern as for Beijing, with a good comparison before 2007 and a larger summer/winter difference in the ATSR data there-after, resulting in increased discrepancy between the ATSR and AVHRR data sets. However, the comparison with
15 MODIS shows an offset between ATSR and MODIS in the years 2004-2007 which seems to be due to larger overestimation of the MODIS AOD at the XiangHe site than at the Beijing site. In some other years the ATSR/MODIS difference appears to be higher in XiangHe than in Beijing, due to higher MODIS AOD. The comparison between AERONET and MODIS AOD data in Figures 4 and 5 show that the difference between the AOD measured at both sites is smaller for AERONET than that for MODIS and hence this may be a MODIS retrieval issue.

20 As the statistics marked in A2 and A3, the data volume of collocated matchups with ground-based reference data may lead to biased validation results and time series comparison. Therefore, pixel by pixel comparison with reference data is necessary. The seasonally averaged AOD and AOD difference maps provide information on not only seasonal but also pixel by pixel comparison. The comparison with the ground-based reference data shows that MODIS provides good results for AOD up to 1.3 in all seasons (see A1). As discussed above, AVHRR failed to retrieve AOD larger than 0.6 for most cases leading to big
25 differences with ATSR and MODIS in the SE of the study area in the summer. However, the AOD patterns of ATSR and AVHRR in other seasons are similar. ATSR has the ability to provide successful retrievals of high AOD, but the small swath limits the spatial coverage. Fig. 9c shows that ATSR and MODIS retrieved AOD are very similar, whereas the difference between AVHRR and MODIS AOD is less than -0.2 for most of the SE part of the study area.

The BEM AOD data set provides a multi-decadal AOD time series from the 1980s to the present, with good quality as
30 evidenced by the comparison with AERONET and MODIS AOD in Fig. 8. The comparison of AVHRR and ATSR-retrieved AOD with the BEM data presented above shows the good agreement of the satellite and ground-based data. However, the AVHRR AOD missed most of the AOD values larger than 0.6, occurring mainly during the summer, but follows BEM AOD well in other seasons and has similar annual patterns. Thus, we conclude that the AOD retrieved from AVHRR can be used to

extend the ATSR data set to before 1995, except in situations with high AOD (>0.6). This conclusion is based on the comparison with only one BEM station. There are at least ten BEM stations in China, as mentioned in Xu et al., (2015). Extension of the AVHRR AOD data set to all China and the comparison with other BEM stations is planned as future work.

In conclusion, the possible combination of the AVHRR and ATSR CDRs, possibly also including MODIS, into a multi-decadal time series is not straightforward. The MODIS performance is better that that of ATSR and AVHRR, but the latter two sensors go back in time before MODIS and hence provide a unique source of information. In addition, none of the satellite sensors performs better than the others at any time and at any location and their combination taking into account the strengths and weaknesses of each of them, may result in a more significant CDR than each of them individually. As shown in this paper, the ADL algorithm does not successfully retrieve AOD for values larger than 0.6, whereas ATSR compares favourably with the reference data for AOD up to about 1.3. On the other hand, the ADV algorithm does not provide good results for latitudes north of 41°, where ADL shows patterns similar to those from MODIS. Furthermore, the comparison shows that before 2007 the AVHRR and ATSR compare quite well and could be used for a meaningful extension of the AOD data records to the early 1980s. However, this conclusion does not apply to the northern sub-region where the surface conditions seem unfavourable for AOD retrieval from ATSR using ADV.

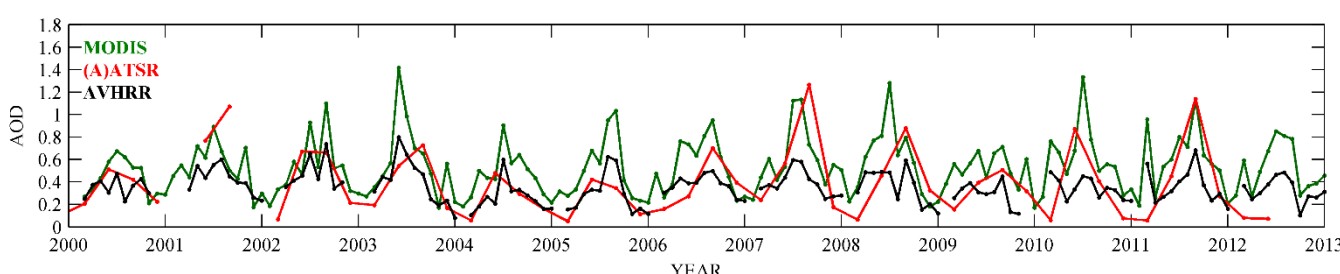

**Figure 10: Comparison of AOD time series over the Beijing AERONET site retrieved from the satellite sensors AVHRR, MODIS and (A)ATSR, for the overlapping years 2000-2015. Note that for AVHRR and MODIS monthly mean AOD data are plotted, while for (A)ATSR they are seasonal means which may provide a shift in the extreme values.**

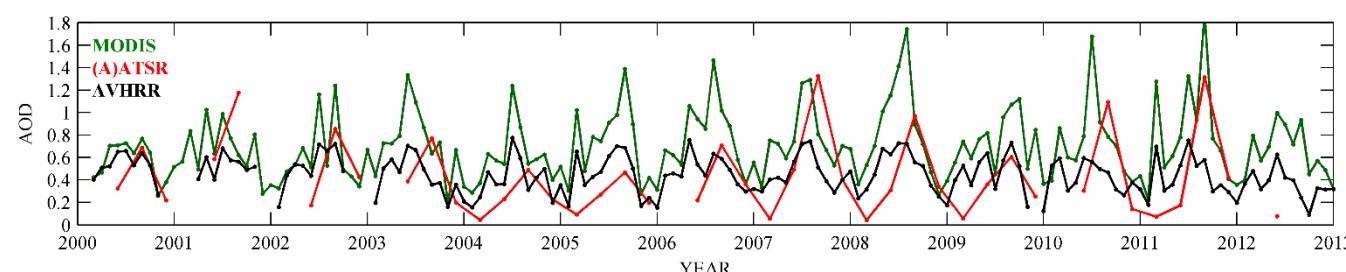

**Figure 11: As Fig. 10, over the XiangHe site.**

*Data availability*. The MODIS data are publicly available from LAADS DAAC website at https://ladsweb.modaps.eosdis.nasa.gov/ (last access: 27 June 2019). The ATSR data are released officially by CCI website at

http://www.icare.univ-lille1.fr/cci (last access: 27 June 2019). The AERONET data are downloaded freely from https://aeronet.gsfc.nasa.gov/ (last access: 27 June 2019).

*Author contributions.* YC and YX produced and analysed the AVHRR AOD data. GL produced and analysed the ATSR AOD data. YC and JG downloaded and analysed the MODIS and AERONET AOD data. LS performed the broadband solar radiation
measurements and provide the BEM AOD data. HC performed the CIMEL sun photometers measurements and provide the CARSNET data. YC, JG, and GL prepared the article with contributions from the other authors. All authors contributed to discussion and interpretation.

*Competing interests.* The authors declare that they have no conflict of interest.

*Financial support.* This research has been supported in part by the Strategic Priority Research Program of the Chinese
Academy of Sciences (grant no. XDA19080303), the Ministry of Science and Technology (MOST) of China under grant nos. 2016YFC0200500 and 2010CB950803, and the National Natural Science Foundation of China (grant nos. 41471306, 41711530127, 41871260, and 41471303), and the Academy of Finland, Research Council for Natural Sciences and Engineering, for financial support (project number: 308295).

*Review statement.* This paper was edited by Thomas Eck and reviewed by Andrew Sayer and other two anonymous referees.

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

## Appendix A

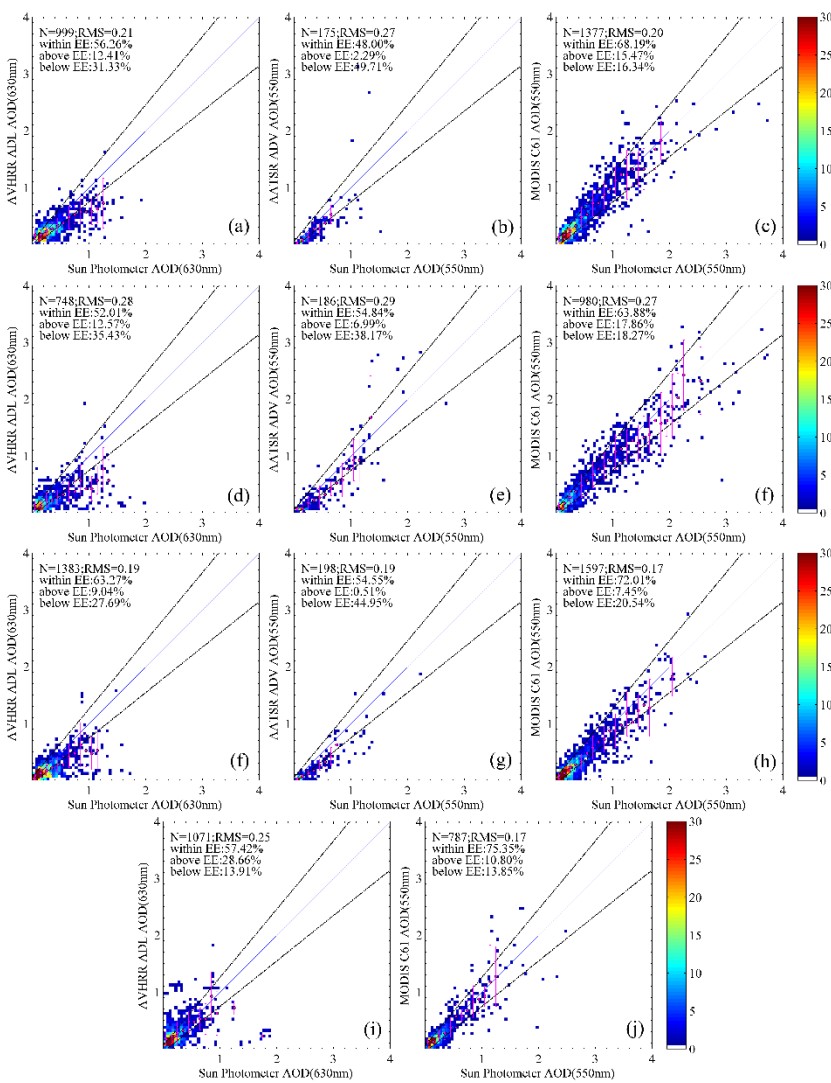

**Figure A1: Seasonal scatter density plots of satellite AOD versus sun photometer AOD. (a), (d), (f), and (i) are AVHRR AOD vs. AERONET AOD in Spring, Summer, Autumn, and Winter; (b), (e), and (g) are ATSR AOD vs. AERONET AOD in Spring, Summer, and Autumn; (c), (f), (h), and (j) are MODIS AOD vs. AERONET AOD in Spring, Summer, Autumn, and Winter.**

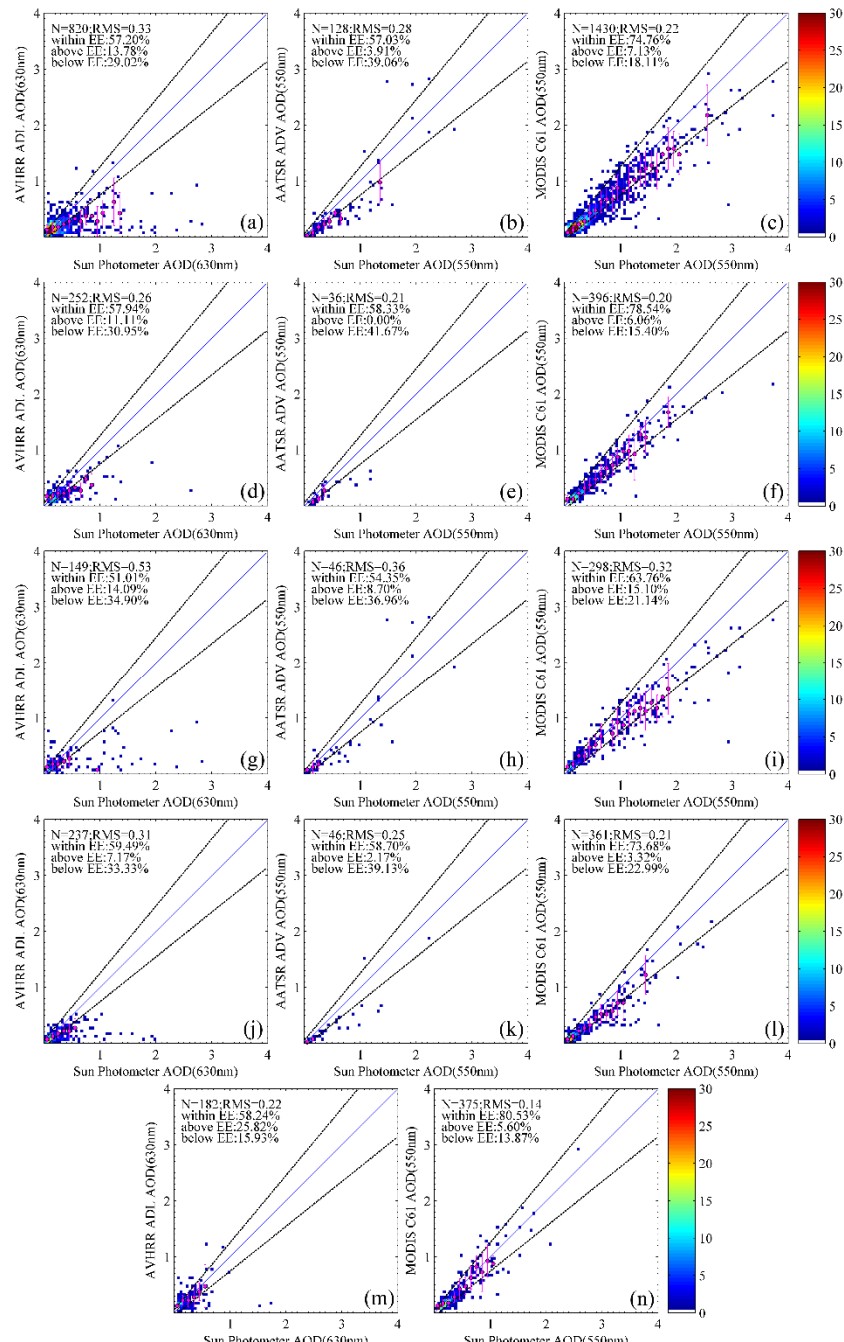

**Figure A2: Scatter density plots of satellite-retrieved AOD versus sun photometer retrieved AOD data from Beijing station. ADL all data (a), in Spring (d), in Summer (g), in Autumn (j), and in Winter (m) vs. AERONET, ADV all data (b) , in Spring (e), in Summer (h), and in Autumn (k) vs. AERONET, MODIS all data (c), in Spring (f), in Summer (i), in Autumn (l), and in Winter (n) vs. AERONET.**

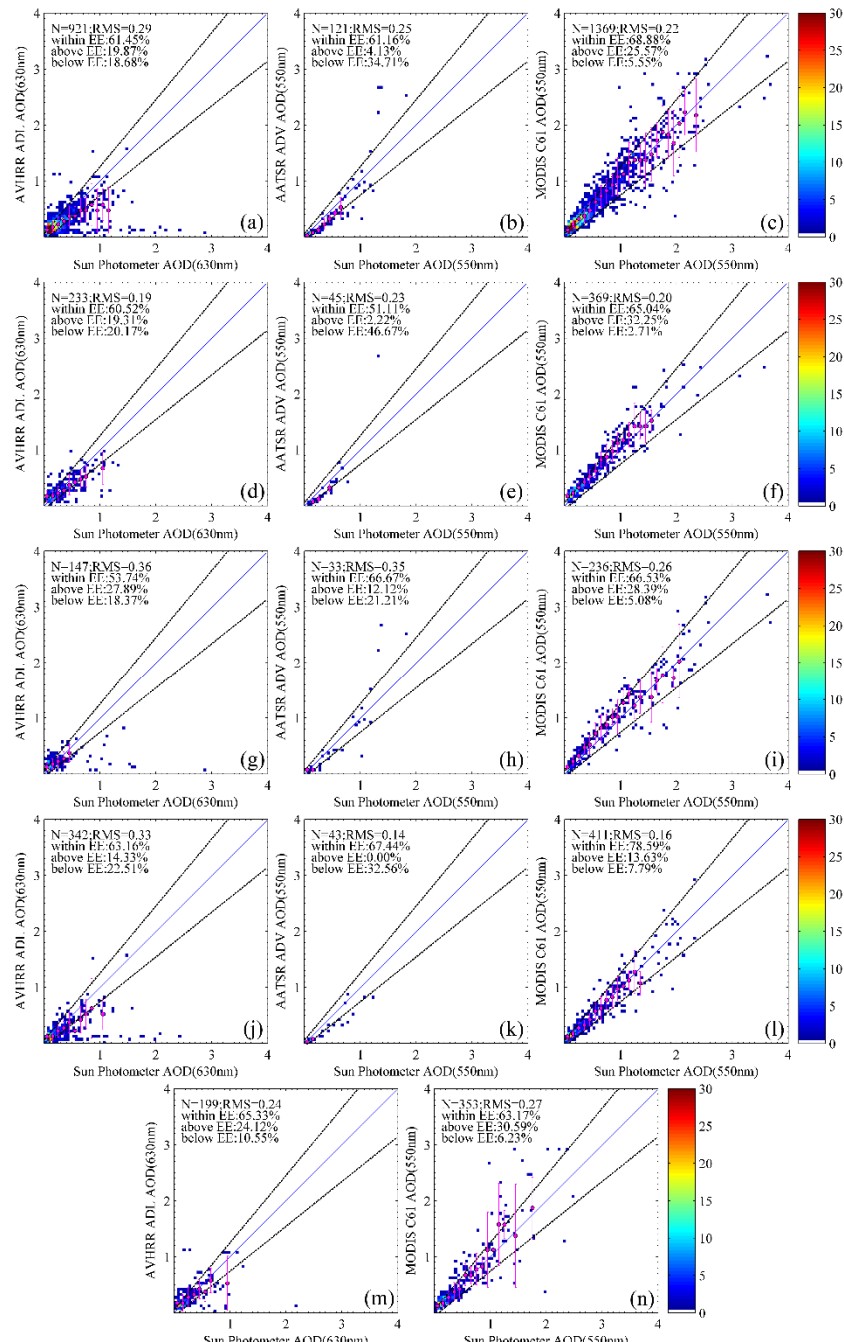

**Figure A3: Scatter density plots of satellite-retrieved AOD versus sun photometer retrieved AOD data from XiangHe station. ADL all data (a), in Spring (d), in Summer (g), in Autumn (j), and in Winter (m) vs. AERONET, ADV all data (b) , in Spring (e), in Summer (h), and in Autumn (k) vs. AERONET, MODIS all data (c), in Spring (f), in Summer (i), in Autumn (l), and in Winter (n) vs. AERONET.**