# Peer review of "Investigations into the Development of a Satellite-Based Aerosol Climate Data Record using ATSR-2, AATSR and AVHRR data over North-Eastern China from 1987 to 2012"

_Atmospheric Measurement Techniques, 2019_

## Referee Comment (RC1) · Andrew Sayer (Referee) · 20 Feb 2019

Andrew Sayer (Referee)

andrew.sayer@nasa.gov

I am writing this review under my own name (Andrew Sayer) as I have worked with all the satellite instruments the authors use here (although on different algorithms for the ATSRs/AVHRRs), am one of the people who developed the MODIS C6.1 aerosol products used, and am familiar with the authors' work on this topic from their previous studies and discussions at conferences.

This paper seeks to assess whether the gap in aerosol optical depth (AOD) data between the end of the AATSR record in 2012 and start of the SLSTR mission in 2016 (using an algorithm developed by co-author de Leeuw's group) can be bridged with

an AVHRR-based algorithm developed by the other authors. Since we are now at the stage where we have multiple satellite records of decent length and quality, it is reasonable to ask how these might be best combined and what sort of consistency can be achieved. So on that front the study is relevant and important. The authors focus on north-eastern China, and in particular two AERONET sites in the Beijing area.

In all honesty the choice of AVHRR and the Beijing region to attempt to bridge the AATSR/SLSTR records seems to be guided in part by the authors' own created data sets and home institution, as the AVHRR sensor is perhaps the least capable of satellite instruments which could have been chosen to fill in this gap, and aerosols in China are complicated so perhaps not the place in the world you'd want to start developing a framework for combining records. The concepts behind the ATSR and AVHRR AOD retrieval algorithms used, and their sampling limitations, are also quite different, which would also affect the expected level of consistency. For example you might expect a MISR-based algorithm to be a better bridge because MISR and the ATSRs are all multiangle, narrow-swath sensors. I suppose I am left asking: why did the authors try to answer question "how well can I bridge the AOD gap between AATSR and SLSTR in north-eastern China using this AVHRR data set?" rather than the more general questions "how well do available satellite records allow us to create a long-term AOD record over north-eastern China?" or "how should we combine different satellite aerosol records?". This analysis is a step in that direction which can provide input to answering those general questions, so I think it does have value. I can understand some scientific rationale behind this because as the authors point out the AVHRRs are the longest available sensor series (going back to around 1980), and that part of the world has undergone a lot of change in that period with considerable aerosol sources. So scientifically it makes some sense to look at this region and sensor. However I would have liked to see that aspect emphasised more in the paper: the title implies a bigger scope, and the bulk of the analysis in the paper sticks to the post-2000 era.

My overall recommendation is for major revisions; I would like to review the revised

version. There is value in this analysis but I think some things need clarification, some need extension, and more big-picture guidance about the insights the authors have got would be useful for readers planning follow-on studies on this topic. Some specific comments and suggestions relating to these general comments are below:

Title: I suggest adding "over north-eastern China" or similar after "Climate Data Record", to better fit the scope and content of the paper. The current title implies a more large-scale analysis.

Introduction: SLSTR was launched in 2016 and this paper is largely about bridging the 2012-2016 gap. However, no SLSTR data are shown in the study. Somewhere in here, could the authors add a few statements about the current status of SLSTR AOD data? If there is a SLSTR product in development, it would make sense to wait until that can be included in the analysis (even if it is preliminary). That way we could answer the question of whether AVHRR can bridge the gap, by seeing how consistent the results are with actual SLSTR retrievals. If there is no SLSTR product in development then ok, but this should be stated, and the premise of the paper becomes more questionable (why try to bridge if there is nothing at the other end of the bridge?). On a related note, the AVHRR data processed here end in 2014 (although the AVHRRs are still flying) so would not be able to bridge the gap anyway unless the authors extend the processing period.

Introduction: As noted in my general comments, I'd like to see more discussion of the bigger picture of why we want to combine satellite records, how it has been done, what the challenges are, and why these sensors and this region were chosen as an example.

Page 6 line 4: I suggest citing a paper here (there are lots of AVHRR aerosol/cloud/surface papers which give instrument descriptions) rather than an ESA webpage, especially for a non-ESA sensor.

Page 6, lines 15-25: Surface reflectance modelling is one of the largest error sources for AOD retrieval, especially for a sensor like AVHRR which has only a few, and quite

broad, spectral bands. The authors' algorithm relates 3.75 micron to 0.64 micron reflectance using an empirical relationship which is a function of surface type. The surface cover type used is obtained from the MODIS MCD12C1 product. This is static for a given location, therefore, if the surface cover changes during the period, the AOD retrieval error will change through time. This is not discussed in the paper but is quite important if the goal is to make a data record going back to the 1980s, as we know there has been a lot of industrialisation since then. Basically we know that surface type has changed over parts of this region, so we know that this key algorithm assumption has been violated. Yet I did not see this discussed in this paper, or in the previous algorithm papers cited here. There should be some discussion and some quantification of the effects of an incorrect surface type classification on the retrieved AOD. (For example, if you use the model for type A instead of type B, how much does the retrieved AOD change, and how systematic is that?) It is also not clear that this surface model can account for the (non-negligible) differences in the spectral response functions between the different sensors, which could result in systematically different reflectances observed for the same underlying surfaces. If these aren't addressed then it undercuts the idea of trying to get a long term data record.

Page 6 line 30: NOAA15 says 199-2002, I guess this should be 1998 or 1999 rather than 199? Also, TETOP-A should read METOP-A.

Page 8 line 14 and later: The authors sometimes refer to "radiance-derived AOD", sometimes "solar radiation", and sometimes "broadband extinction", when describing one of the ground-based data sets. These phrases, so far as I can tell, all refer to the same data set, but they do not have any words in common. I suggest keeping the terminology consistent to make things clear. My personal preference would be to say "broadband solar radiation" but I'd leave that up to the authors. This refers to some figure captions as well as the body text.

Page 9 line 5: There is now an AERONET version 3 paper published, which could be cited here. https://www.atmos-meas-tech.net/12/169/2019/

Page 10 line 21: This says that the AVHRR retrievals are available from 1983-2014. But from page 6, only data back to 1987 are used in this study. Why not include the first four years too, since the authors state they processed the data?

Figure 2: We know that AOD changes a lot in different seasons. We also know that satellite biases are affected by geometry, AOD/aerosol type, and surface reflectance, which also change a lot in different seasons. We also know that sampling is related to cloud and snow cover, which also change a lot in different seasons. We know that all these factors can affect different algorithms in different ways. Given that, I do not find multiannual means as presented in this Figure to be useful, as we have no idea which factors are contributing to the differences observed here. I suggest changing this to show multiannual seasonal composites instead of overall multiannual means. I feel that will be more useful and allow for more insight about these confounding factors. Otherwise the authors need to provide a justification why presenting the data in this way is useful, given these issues.

Page 12, line 23: should be changed to "to MODIS Dark Target over land", for completeness. This expression does not apply to the Deep Blue or merged product (which is what is actually used here).

Page 13, lines 28-30: If my reading is correct, the authors are saying that the MODIS C6.1 validation results obtained here have different error characteristics than the MODIS C6.1 validation results over the same region obtained by Sogacheva et al (2018). Is that correct? If so, it seems surprising, so needs to be examined in more detail. Is there an error in one of the studies? Where does the difference come from? If it is a result of a different sampling period or sites, that is important to discuss, as it indicates that the regional validation is not robust and therefore that we cannot take these results as representative. This should be clarified in the text.

Figures 4-6 (and 9, 10): I find these a little hard to interpret. We are trying to see how consistent the data sets are with AERONET and each other, yet by plotting all of them

on separate panels it makes harder to see how consistent they are. I suggest plotting all the data sets on a single panel instead, so we can directly compare them. I realise that the ATSR data shown are seasonal while the others are monthly, due to sampling limitations. In that case one option is just to plot everything (including AERONET) as a seasonal rather than monthly time series. This would also remove the issue of different expected extrema which the authors mention in captions.

Figures 7, 8: these are interesting, but I feel they should be expanded to make the paper more complete. They show that the monthly solar broadband-based AOD is a reasonable proxy for AOD from AERONET or MODIS. But why not also include scatter plots like figure 7 for AVHRR and AATSR? And why not include the full time series, as well as ATSR data, in figure 8? Basically the paper is about a 1983/7 and onward AOD data set over Beijing, but nowhere is a full time series of these data actually shown in the paper.

Conclusions: The paper also builds on other recent work by the de Leeuw group which assessed a combination of the ATSR and MODIS sensors to get a longer-term AOD over China. This is mentioned a few times early on, but I think the conclusion should be expanded to put these studies in perspective with each other, and give some more specific thoughts/suggestions on how best to combine data records. Overall this paper has some good points but suffers from not always following through to give the bigger picture (i.e. when you have finished your investigation of the long-term climate data record, as promised in the title, what does it actually look like)?

---

## Referee Comment (RC2) · Anonymous Referee #2 · 28 Feb 2019

Review of "Investigations into the Development of a Satellite-Based Aerosol Climate Data Record using ATSR-2, AATSR and AVHRR data" for Atmospheric Measurement Techniques. This paper try to discuss the feasibility of using AVHRR to continue the aerosol optical depth (AOD) record from AATSR ending in 2012 to SLSTR starting at 2016 over Beijing-Tianjing-Hebei region. The study is relevant and the potential product will benefit the aerosol community. However, there are some major issues that need to be addressed before it is suitable for publishing.

1. The reason author chooses to use AVHRR is because not only can this data bridge the gap between AATSR and SLSTR but also it can extend the data record to 1983.

[Figure]

This idea is presented in introduction, but there is only one plot Figure 8 shows the AVHRR data before 2000. All other analyses are focused between 2000 to 2012. I think if we only consider this 2000-2012 period of time, there are a lot more aerosol products that can be used with much lower uncertainties. Thus, more analyses are needed to understand AVHRR through the entire data record or empirically correct AVHRR data to make it more suitable for a long term data record. 2. This is a paper about continue data from AATSR to SLSTR. But I didn't see a session in data talking about SLSTR. 3. The author uses a very large portion of paper introducing aerosols and their facts. It is really not to the point of this article. Please make the introduction more concise. 4. To me it makes more sense to validate the radiance retrieved AOD against AERONET. Or maybe against MODIS to show the sampling bias. Then rely on radiance retrieved AOD to validate everything else consistently from 1983 to current. 5. The title doesn't indicate the study region.

---

## Referee Comment (RC3) · Anonymous Referee #3 · 16 Mar 2019

This manuscript compares aerosol retrievals from 3 different sensors (MODIS, ATSR, and AVHRR) with ground based validation data (including AERONET) over a 10x10 degree box centered on eastern China. The goal is a worthy one, the creation of a continuous climate data record of satellite-retrieved aerosol optical depth from the early 1980's until the present day period. I believe this work can be published if the authors are willing to substantially change the manuscript.

Most of my comments are embedded within the PDF attached, but I will summarize a few points.

1) As was mentioned by the other two reviewers, where is the SLSTR data? If this data

is not available, this manuscript can still add value, but not much (in its current form).

2) MISR shares a lot of similarities to ATSR including: swath size, multi-angle viewing, equatorial crossing time, and algorithm heritage (over-land). Additionally, I expect that the error statistics for MISR (over this region) are quite a bit better than for any other sensor used in this paper. In fact, given MISR's very long data record (2000–>2019 and counting), its similarities with ATSR, its overlap with *both* ATSRs *and* SLSTR, I think it makes much more sense to use MISR to stitch together the ATSRs and SLSTR. Once those two datasets are harmonized with MISR (globally, not for one region), I would then look back and compare with AVHRR (globally, or at least using all regions available).

3) Please find a way to make this work much more global. A 10x10 degree region is not a very useful climate data record, especially in a region with so much dust and pollution transport. Additionally, the authors could show consistencies and discrepancies with other sensors via a map of gridded correlations and differences (using seasonal AODs, compared with other sensors).

4) As a third party (I work with MISR data) with no stake in any of these instruments (at least data from the ones presented), it seems pretty clear from this small dataset that MODIS provides the best available AOD here (by far). One (or more) of three things is going on here: (1) AATSR's aerosol retrieval algorithm is inferior to MODIS, (2) AATSR's sample size in this region is so small as to border on the irrelevant, or (3) the region selected is so small that regional biases in the algorithm dominate your observed errors.

If (1), I have to wonder why bother stitching together AOD from ATSR and AATSR with SLSTR (and AVHRR) at all? Even though ATSR, AATSR, and SLSTR all lack a blue band (which will significantly degrade performance over brighter regions), this should be compensated by the additional view angle. If the current algorithm is insufficient, maybe a new one should be developed. Otherwise, if MODIS truly gives better performance, just create the CDR using MODIS, AVHRR, and VIIRS, which would be easier anyways. If (2) or (3) see point 3) above, you need more data.

I don't want to discourage the authors, this work does have the potential to add value, but more work needs to be done.

Please also note the supplement to this comment:
https://www.atmos-meas-tech-discuss.net/amt-2019-26/amt-2019-26-RC3-supplement.pdf

**Supplement:**

[revised manuscript text omitted]

20    emitted by natural processes such as the interaction between wind and waves which produces sea spray aerosol in quantities and composition which primarily depend on wind speed (e.g., O'Dowd and de Leeuw, 2007, de Leeuw et al., 2011), entrainment of dust into the atmosphere by the action of the wind producing desert dust aerosol (e.g., Yu et al., 2018, Alizadeh-Choobari et al., 2018), forest fires producing biomass burning aerosol (e.g., Miller et al., 2011, Kaiser et al. 2012), or volcanic eruptions producing volcanic ash (e.g. Lu et al., 2016). Wind-blown dust can also be of anthropogenic origin

25    such as from agricultural or construction activities (e.g. Gillette, 1988) and biomass burning aerosol can be produced by anthropogenic activities such as straw burning (e.g. Zhang et al., 2016, Chen et al., 2018) or man-induced forest fires (Chen et al., 2018). Aerosols such as black carbon, produced from incomplete combustion, play an important role in atmospheric processes due to the absorption of solar light which affects the evolution of the atmospheric boundary layer and thus vertical mixing of air pollution. Precursor gases can be of both natural and anthropogenic origin. $NO_2$ is mainly formed from

30    combustion processes but also natural sources contribute such as lightning (Stark et al., 1996) and soil emissions (e.g., Oertel et al., 2016). The $SO_2$ emissions from power plants have been largely reduced since the second half of the 20th century and over most of the world the concentrations are low (e.g. Fioletov et al., 2016), the occurrence of high $SO_2$ concentrations in

the atmosphere is used as a strong indicator of volcanic activity (e.g. Theys et al., 2013, Carn et al. 2017). Volatile organic compounds (VOCs) are emitted by plants and from anthropogenic sources and are nowadays identified as strong contributors to aerosol concentrations (e.g. Bai et al., 2018, Stavrakou et al., 2016).

In addition to this large variety of sources, the aerosol concentrations vary strongly with meteorological conditions, which in

5  turn are affected by large scale synoptic conditions and weather systems (Li et al., 2017a; Miao et al., 2017; Yim et al., 2019). The aerosol concentrations also vary with economic development such as industrialization, urbanization and the ensuing policy measures to reduce adverse effects on health and climate. Since the industrial revolution in the 18th century, the emission of air pollutants has been increasing until adverse effects were recognized and measures were developed and implemented to reduce emissions and concentrations of pollutants (e.g., Brimblecombe, 2006). In particular, in the second

10  half of the 20th century the effects of $SO_2$ on forests and lakes, also known as acid rain, was recognized and measures were taken to reduce the $SO_2$ emissions  Later, the adverse effects of $NO_2$ emissions and effects of aerosols, especially fine particulate matter, on air quality, health and climate were recognized and reduced. These led to the reduction of air pollution in developed countries, in particular in North America and Europe (Guerreiro et al., 2014), but in developing countries with increasing industrial activity and urbanization the concentrations continued to increase (Hao et al., 2000).

15  Examples are China and India where the concentrations of pollutants are amongst the highest in the world. Taking China as an example, recent publications show the effect of policy measures on the reduction of the most polluting trace gases $SO_2$ and $NO_2$ (van der A et al., 2017), which, as precursor gases, also affect the concentrations of aerosols. In particular, the emissions of $SO_2$ were reduced as part of the 11th Five-Year Plan (2006-2010) (Zheng et al., 2018), but the emissions of $NO_2$ continued to increase (e.g., van der A et al., 2017) until the 12th Five-Year Plan (2011-2015). Large emission

20  reductions were achieved after 2013 when the Clean Air Action was enacted and implemented and the $NO_2$ concentrations decreased (Zheng et al., 2018). Starting from 2011, aerosol concentrations decreased in China as shown, e.g., from satellite observations of the aerosol optical depth (AOD) (Zhang et al., 2017, Zhao et al., 2017, de Leeuw et al, 2018, Sogacheva et al., 2018b).

Observations of the concentrations of trace gases and aerosols in China are publicly available since several observational

25  networks have been established, such as CARE-China (Xin et al., 2015), and the NASA's AERONET (AErosol RObotic NETwork) (Holben et al. 1998) with observations mainly in the east of China, the Chinese CARSNET (Chinese Aerosol Remote Sensing Network) (Che et al., 2009; 2015) and SONET (Sun-sky radiometer Observation NETwork) (Li et al., 2018), all of which provide data across the whole country. However, most of these observations were established in the last decade and very few, if any, historical data on large scale are available for the construction of the long time series needed to

30  show the evolution of pollutant concentrations over many years and analyse the effects of different contributions. Here, satellite data may offer a solution. The most common satellites used for the observation of trace gases and aerosols offer information since the beginning of the 21st century and, by combining the information from different instruments, time series encompassing two decades can be constructed (de Leeuw et al., 2018, Sogacheva et al., 2018b). Satellite information has been used together with model simulation to analyse the effects of natural and anthropogenic contributions on the

concentrations of trace gases and aerosols (Kang et al., 2018). In another study combining satellite data with ground-based observations, the role of precursor gases, in particular VOCs, and photochemical reactions in the formation of aerosols (PM$_{2.5}$) was revealed (Bai et al., 2018).

In this study we focus on aerosols, and in particular on the AOD observed by satellites. The most common instrument used for aerosol retrieval is the Moderate Resolution Imaging Spectroradiometer (MODIS) which was launched on the Terra satellite, in a morning orbit with equator crossing time (descending) at 10:30 local time (LT), in December 1999 and on the Aqua satellite (ascending, equator crossing time 13:30 LT) in May 2002. MODIS thus provides an AOD time series since 2000 and is still operational. The Along Track Scanning Radiometer ATSR-2, a dual view instrument, was launched in 1995 on the European Space Agency (ESA) satellite ERS-2 and provided data until 2003. The Advanced ATSR (AATSR) is a similar instrument launched in 2002 on the ESA platform ENVISAT, which was lost in April 2012. The AOD over China from ATSR-2 and AATSR is consistent (Sogacheva et al., 2018a) and hence, together these instruments provide a 17-year AOD time series, 1995-2012 (Popp et al., 2016, de Leeuw et al., 2018). Combining ATSR and MODIS, 22-year AOD measurements were constructed, showing the AOD increased until about 2006, and then clear decreased since 2011 over China (de Leeuw et al., 2018, Sogacheva et al., 2018b). MODIS is approaching the end of its life time, but the AOD time series will be extended with NPP/VIIRS (cf. Levy et al., 2015). The AATSR time series will be extended with data from the operational Sea and Land Surface Temperature Radiometer (SLSTR) which was launched on Sentinel-3 in February 2016. This leaves a gap of about 4 years between AATSR and SLSTR. This gap could be filled with MODIS data, as described in Sogacheva et al. (2018b) who combined ATSR and MODIS/Terra (C6.1) data to construct an AOD time series 1995-2017. The objective of the current study is to investigate whether this can also be done by using AVHRR data, using the AOD data set over land which was recently presented by Xue et al. (2017). The advantage of using AVHRR data is the long time series which would allow extension back to the earliest AVHRR AOD retrievals available, in this case 1983, and thus construct a time series 1983-present. The Xue et al. (2017) data set encompasses only two relatively small areas over Europe and China, but covers the complete period, as opposed to the AVHRR global over land AOD data set recently presented by Hsu et al. (2017) and Sayer et al. (2017) which encompasses several distinct time periods. Hence we focus here on the Xue et al (2017) AVHRR AOD data set over China and compare that with ATSR-derived AOD data to determine its suitability for merging (as done for ATSR/MODIS by Sogacheva et al. 2018b), and thus extending both before the ATSR-2 era and after the AATSR era until SLSTR. In this study we use both ground-based reference data, from AERONET (Holben et al., 1988) and from CARSNET (Che et al, 2009, 2015), and MODIS C6.1 AOD data for comparison and evaluation. Reference data and MODIS/Terra data for inter-comparison are not available for the period before 2000 and therefore we also use radiance - derived AOD data (Xu et al., 2015, Guo et al, 2016b). Data sets and methods used are presented in section 2. An overview of the data and an evaluation of their quality are presented in section 3, including a comparison of the various data sets. The results are discussed and conclusions are presented in section 4.

[Figure]

**2 Method**

**2.1 Study area**

The study area is located over East China, i.e. between 110 ° and 120 ° E and 35 ° and 45 °N (Fig. 1), which is divided into two sub-regions by the Taihang Mountain range with the North China Plain (NCP) and large urban agglomerations like

5    Beijing and Tianjin and the Hebei province (together BTH) to the SE and the mountainous terrain to the NW extending over the Loess Plateau in Shanxi Province and the Inner Mongolia plateau. The Taihang mountain range forms a natural barrier for the transport of air pollution resulting in the frequent accumulation of pollutants and the occurrence of haze over the BTH area and the NCP (e.g., Sundström et al., 2012; Wang et al., 2013). The satellite-derived AOD maps in Fig. 2 show that this line also roughly divides high AOD in the SW of the study area and low AOD in the NW. The background in Fig. 1 is a land

10   cover map showing that the major land cover types in the study area are croplands in the SE and grassland to the NW, which are intersected by mixed forest and closed shrublands as shown in the inset in Fig. 1.

[Figure]

**Figure 1: Study area, with the locations of the ground-based reference sites discussed in Sect. 2.3 (CARSNET: red squares, AERONET: blue circles, solar radiation station: green triangle) overlaid on the IGBP land cover map. Land cover is colour coded,**
15   **see the legend to the right of the map: Water (0), Evergreen Needleleaf forest (1), Evergreen Broadleaf forest (2), Deciduous Needleleaf forest (3), Deciduous Broadleaf forest (4), Mixed forest (5), Closed shrublands (6), Open shrublands (7), Woody savannas (8), Savannas (9), Grasslands (10), Permanent wetlands (11), Croplands (12), Urban and built-up (13), Cropland/Natural vegetation mosaic (14), Snow and ice (15) and Barren or sparsely vegetated (16).**

[revised manuscript text omitted]

de Leeuw, G., Andreas, E. L., Anguelova, M. D., Fairall, C. W., Lewis, E. R., O'Dowd, C., Schulz, M., and Schwartz, S. E.: Production flux of sea spray aerosol, Rev. Geophys., 49, doi:	10.1029/2010RG000349, 2011.

de Leeuw, G., Holzer-Popp, T., Bevan, S., Davies, W. H., Descloitres, J., Grainger, R. G., Griesfeller, J., Heckel, A., Kinne, S., Klüser, L., Kolmonen, P., Litvinov, P., Martynenko, D., North, P., Ovigneur, B., Pascal, N., Poulsen, C., Ramon, D., Schulz, M., Siddans, R., Sogacheva, L., Tanré, D., Thomas, G. E., Virtanen, T. H., von Hoyningen-Huene, W., Vountas, M., and Pinnock, S.: Evaluation of seven European aerosol optical depth retrieval algorithms for climate analysis, Remote Sens. Environ., 162, 295-315, doi:10.1016/j.rse.2013.04.023, 2015.

de Leeuw, G., Sogacheva, L., Rodriguez, E., Kourtidis, K., Georgoulias, A. K., Alexandri, G., Amiridis, V., Proestakis, E., Marinou, E., Xue, Y., van der A, R.: Two decades of satellite observations of AOD over mainland China using ATSR-2, AATSR and MODIS/Terra: data set evaluation and large-scale patterns, Atmos. Chem. Phys., 18, 1573-1592, doi: 10.5194/acp-18-1573-2018, 2018.

Eck, T. F., Holben, B. N., Reid, J. S., Dubovik, O., Smirnov, A., O'Neill, N. T., Slutsker, I., and Kinne, S.: Wavelength dependence of the optical depth of biomass burning, urban, and desert dust aerosols, J. Geophys. Res.-Atmos., 104, 31333–31349, doi: 10.1029/1999JD900923, 1999.

Flowerdew, R. J., and Haigh, J. D.: An approximation to improve accuracy in the derivation of surface reflectance from multi-look satellite radiometers, Geophys. Res. Lett., 23, 1693–1696, doi: 10.1029/95GL01662, 1995.

Fioletov, V. E., McLinden, C. A., Krotkov, N., Li, C., Joiner, J., Theys, N., Carn, S., and Moran, M. D.: A global catalogue of large SO2 sources and emissions derived from the Ozone Monitoring Instrument, Atmos. Chem. Phys., 16, 11497-11519, doi: 10.5194/acp-16-11497-2016, 2016.

Gilette, D.A.: Threshold Friction Velocities for Dust Production for Agricultural Soils. J. Geophys. Res.-Atmos., 93, 12645–12662, doi: 10.1029/JD093iD10p12645, 1988.

[revised manuscript text omitted]

Zhang. L., Liu, Y. and Hao, L.: Contributions of open crop straw burning emissions to PM2.5 concentrations in China. Environ. Res. Lett., 11, 014014, doi:10.1088/1748-9326/11/1/014014, 2016.

[Figure]

Zhang, J., Reid, J.S., Alfaro-Contreras, R., and Xian, P.: Has China been exporting less particulate air pollution over the past decade? Geophys. Res. Lett., 44, 2941–2948, doi:10.1002/2017GL072617, 2017.

Zhao, T. X.-P., Laszlo, I., Guo, W., Heidinger, A., Cao, C., Jelenak, A., Tarpley, D., and Sullivan, J.: Study of long-term trend in aerosol optical thickness observed from operational AVHRR satellite instrument, J. Geophys. Res., 113, D07201, doi:10.1029/2007JD009061, 2008.

Zhao, B., Jiang, J.H., Gu, Y., Diner, D., Worden, J., Liou, K.-N., Su, H., Xing, J., Garay, M., and Huang, L.: Decadal-scale trends in regional aerosol particle properties and their linkage to emission changes, Environ. Res. Lett., 12, 054021(2017), doi.org/10.1088/1748-9326/aa6cb2, 2017.

Zheng, B., Tong, D., Li, M., Liu, F., Hong, C., Geng, G., Li, H., Li, X., Peng, L., Qi, J., Yan, L., Zhang, Y., Zhao, H., Zheng, Y., He, K., and Zhang, Q.: Trends in China's anthropogenic emissions since 2010 as the consequence of clean air actions, Atmos. Chem. Phys. Discuss., https://doi.org/10.5194/acp-2018-374, in review, 2018.

[Figure]

[Figure]

**Appendix A**

[Figure]

**Figure A1: Seasonal scatter density plots of satellite AOD versus sun photometer AOD. (a), (d), (f), and (i) are AVHRR AOD vs. AERONET AOD in Spring, Summer, Autumn, and Winter; (b), (e), and (g) are ATSR AOD vs. AERONET AOD in Spring, Summer, and Autumn; (c), (f), (h), and (j) are MODIS AOD vs. AERONET AOD in Spring, Summer, Autumn, and Winter.**

[Figure]

[Figure]

[Figure]

**Figure A2: Scatter density plots of satellite-retrieved AOD versus sunphotometer retrieved AOD data from Beijing station. ADL all data (a), in Spring (d), in Summer (g), in Autumn (j), and in Winter (m) vs. AERONET, ADV all data (b), in Spring (e), in Summer (h), and in Autumn (k) vs. AERONET, MODIS all data (c), in Spring (f), in Summer (i), in Autumn (l), and in Winter (n) vs. AERONET.**

[Figure]

[Figure]

[Figure]

**Figure A3: Scatter density plots of satellite-retrieved AOD versus sunphotometer retrieved AOD data from XiangHe station. ADL all data (a), in Spring (d), in Summer (g), in Autumn (j), and in Winter (m) vs. AERONET, ADV all data (b) , in Spring (e), in Summer (h), and in Autumn (k) vs. AERONET, MODIS all data (c), in Spring (f), in Summer (i), in Autumn (l), and in Winter (n) vs. AERONET.**

---

## Author Comment (AC1) · 15 May 2019

**andrew.sayer@nasa.gov**

I am writing this review under my own name (Andrew Sayer) as I have worked with all the satellite instruments the authors use here (although on different algorithms for the ATSRs/AVHRRs), am one of the people who developed the MODIS C6.1 aerosol products used, and am familiar with the authors' work on this topic from their previous studies and discussions at conferences.

This paper seeks to assess whether the gap in aerosol optical depth (AOD) data between the end of the AATSR record in 2012 and start of the SLSTR mission in 2016 (using an algorithm developed by co-author de Leeuw's group) can be bridged with an AVHRR-based algorithm developed by the other authors. Since we are now at the stage where we have multiple satellite records of decent length and quality, it is reasonable to ask how these might be best combined and what sort of consistency can be achieved. So on that front the study is relevant and important. The authors focus on north-eastern China, and in particular two AERONET sites in the Beijing area.

In all honesty the choice of AVHRR and the Beijing region to attempt to bridge the AATSR/SLSTR records seems to be guided in part by the authors' own created data sets and home institution, as the AVHRR sensor is perhaps the least capable of satellite instruments which could have been chosen to fill in this gap, and aerosols in China are complicated so perhaps not the place in the world you'd want to start developing a framework for combining records. The concepts behind the ATSR and AVHRR AOD retrieval algorithms used, and their sampling limitations, are also quite different, which would also affect the expected level of consistency. For example, you might expect a MISR-based algorithm to be a better bridge because MISR and the ATSRs are all multiangle, narrow-swath sensors. I suppose I am left asking: why did the authors try to answer question "how well can I bridge the AOD gap between AATSR and SLSTR in north-eastern China using this AVHRR data set?" rather than the more general questions "how well do available satellite records allow us to create a long-term AOD record over north-eastern China?" or "how should we combine different satellite aerosol records?". This analysis is a step in that direction which can provide input to answering those general questions, so I think it does have value. I can understand some scientific rationale behind this because as the authors point out the AVHRRs are the longest available sensor series (going back to around 1980), and that part of the world has undergone a lot of change in that period with considerable aerosol sources. So scientifically it makes some sense to look at this region and sensor. However I would have liked to see that aspect emphasised more in the paper: the title implies a bigger scope, and the bulk of the analysis in the paper sticks to the post-2000 era.

My overall recommendation is for major revisions; I would like to review the revised version. There is value in this analysis but I think some things need clarification, some need extention, and more

big-picture guidance about the insights the authors have got would be useful for readers planning follow-on studies on this topic. Some specific comments and suggestions relating to these general comments are below:

Re: Thank you for your recommendation and give us a chance to improve the manuscript. We have addressed these points in the revised manuscript.

Title: I suggest adding "over north-eastern China" or similar after "Climate Data Record", to better fit the scope and content of the paper. The current title implies a more large-scale analysis.

Re: We revised the title by adding "over North-Eastern China from 1987 to 2012" to indicate both time and regions.

Introduction: SLSTR was launched in 2016 and this paper is largely about bridging the 2012-2016 gap. However, no SLSTR data are shown in the study. Somewhere in here, could the authors add a few statements about the current status of SLSTR AOD data? If there is a SLSTR product in development, it would make sense to wait until that can be included in the analysis (even if it is preliminary). That way we could answer the question of whether AVHRR can bridge the gap, by seeing how consistent the results are with actual SLSTR retrievals. If there is no SLSTR product in development then ok, but this should be stated, and the premise of the paper becomes more questionable (why try to bridge if there is nothing at the other end of the bridge?). On a related note, the AVHRR data processed here end in 2014 (although the AVHRRs are still flying) so would not be able to bridge the gap anyway unless the authors extend the processing period.

Re: Thanks for this remind. To the current, SLSTR data are not available. Hence, we switched to extend ATSR data set back to 1980s, no longer focus on bridging gap between AATSR and SLSTR.

Introduction: As noted in my general comments, I'd like to see more discussion of the bigger picture of why we want to combine satellite records, how it has been done, what the challenges are, and why these sensors and this region were chosen as an example.

Re: Referee 3 also mentioned this point. We switched focus on making an extention back to 1980s for ATSR data set, no longer pay attention to bridge the gap of AATSR and SLSTR. Corresponding revisions have been made in new version of manuscript.

Page 6 line 4: I suggest citing a paper here (there are lots of AVHRR aerosol/cloud/surface papers which give instrument descriptions) rather than an ESA webpage, especially for a non-ESA sensor.

Re: Here we update the citation to "NOAA KLM user's guide" from NOAA official website.

Page 6, lines 15-25: Surface reflectance modelling is one of the largest error sources for AOD retrieval, especially for a sensor like AVHRR which has only a few, and quite broad, spectral bands. The authors' algorithm relates 3.75 micron to 0.64 micron reflectance using an empirical relationship which is a function of surface type. The surface cover type used is obtained from the MODIS MCD12C1 product. This is static for a given location, therefore, if the surface cover changes during the period, the AOD retrieval error will change through time. This is not discussed in the paper but is quite important if the goal is to make a data record going back to the 1980s, as we know there has been a lot of industrial station since then. Basically we know that surface type has changed over parts of this region, so we know that this key algorithm assumption has been violated.

Yet I did not see this discussed in this paper, or in the previous algorithm papers cited here. There should be some discussion and some quantification of the effects of an incorrect surface type classification on the retrieved AOD. (For example, if you use the model for type A instead of type B, how much does the retrieved AOD change, and how systematic is that?) It is also not clear that this surface model can account for the (non-negligible) differences in the spectral response functions between the different sensors, which could result in systematically different reflectances observed for the same underlying surfaces. If these aren't addressed then it undercuts the idea of trying to get a long term data record.

Re: This version of AVHRR AOD data set does have some limitations or disadvantages. As you know, it is very difficult to retrieval AOD when there only one visible band, besides there is no enough prior knowledge in the early time. This paper focus on the analysis of the first version of AVHRR AOD results but not the algorithm itself. Now we are trying to make more detailed validation and improve this algorithm. However, the newly version of long term AVHRR AOD data set is not available now.

Page 6 line 30: NOAA15 says 199-2002, I guess this should be 1998 or 1999 rather than 199? Also, TETOP-A should read METOP-A.

Re: The data from NOAA-15 is from 1992 to 2002, so it should be 1999. These two typos have been revised in new version of manuscript.

Page 8 line 14 and later: The authors sometimes refer to "radiance-derived AOD", sometimes "solar radiation", and sometimes "broadband extinction", when describing one of the ground-based data sets. These phrases, so far as I can tell, all refer to the same data set, but they do not have any words in common. I suggest keeping the terminology consistent to make things clear. My personal preference would be to say "broadband solar radiation" but I'd leave that up to the authors. This refers to some figure captions as well as the body text.

Re: Here we use "BEM (broadband extinction method) AOD" throughout new manuscript, as it does in Dr. Ling Sun's paper. https://doi.org/10.1016/j.atmosenv.2015.08.042

Page 9 line 5: There is now an AERONET version 3 paper published, which could be cited here. https://www.atmos-meas-tech.net/12/169/2019/

Re: Thanks. We have citied this paper in new version of manuscript.

Page 10 line 21: This says that the AVHRR retrievals are available from 1983-2014. But from page 6, only data back to 1987 are used in this study. Why not include the first four years too, since the authors state they processed the data?

Re: All instruments we selected in this paper are onboard satellites along with descending orbit. For years from 1983 to 1986, AVHRR raw data are not consistent, data only from 1986.11 to 1986.12 (from NOAA-10), 1985.07 to 1985.10 (from NOAA-8), 1984.05 to 1984.06 (from NOAA-8) and 1984.07 (from NOAA-6), and 1983.01 to 1983.03 (from NOAA-6) are available. So we abandon data from 1983 to 1986. This is stated in new version of manuscript.

Figure 2: We know that AOD changes a lot in different seasons. We also know that satellite biases are affected by geometry, AOD/aerosol type, and surface reflectance, which also change a lot in

different seasons. We also know that sampling is related to cloud and snow cover, which also change a lot in different seasons. We know that all these factors can affect different algorithms in different ways. Given that, I do not find multiannual means as presented in this Figure to be useful, as we have no idea which factors are contributing to the differences observed here. I suggest changing this to show multiannual seasonal composites instead of overall multiannual means. I feel that will be more useful and allow for more insight about these confounding factors. Otherwise the authors need to provide a justification why presenting the data in this way is useful, given these issues.

Re: Thanks, seasonal characteristics of differences and spatial distribution are really helpful for understanding AOD retrieval algorithms and data sets. Seasonal AOD and AOD differences have been added into new version of manuscript.

Page 12, line 23: should be changed to "to MODIS Dark Target over land", for completeness. This expression does not apply to the Deep Blue or merged product (which is what is actually used here).

Re: This has been revised in new version of manuscript.

Page 13, lines 28-30: If my reading is correct, the authors are saying that the MODIS C6.1 validation results obtained here have different error characteristics than the MODIS C6.1 validation results over the same region obtained by Sogacheva et al (2018). Is that correct? If so, it seems surprising, so needs to be examined in more detail. Is there an error in one of the studies? Where does the difference come from? If it is a result of a different sampling period or sites, that is important to discuss, as it indicates that the regional validation is not robust and therefore that we cannot take these results as representative. This should be clarified in the text.

Re: Yes, that is correct, but the region is not e exactly the same. Moreover, the difference is that we used data from both AERONET and CARSNET in validation while Sogacheva et al (2018) only used AERONET data in validation. This is why there are different error characteristics. As shown by fig.1 in manuscript, all AERONET sites are in or around Beijing where main surface types are urban, croplands or mixed forest and aerosol sources are mainly anthropogenic, on the contrary CARSNET sites distribute in more regions in study area especially in sparsely vegetated regions where the surface and aerosol circumstances are significantly different. Hence, the conclusions are with small differences. This has been clarified in the new version of manuscript.

Figures 4-6 (and 9, 10): I find these a little hard to interpret. We are trying to see how consistent the data sets are with AERONET and each other, yet by plotting all of them on separate panels it makes harder to see how consistent they are. I suggest plotting all the data sets on a single panel instead, so we can directly compare them. I realize that the ATSR data shown are seasonal while the others are monthly, due to sampling limitations. In that case one option is just to plot everything (including AERONET) as a seasonal rather than monthly time series. This would also remove the issue of different expected extreme which the authors mention in captions.

Re: Seasonal comparison rather than monthly comparison in time series is definitely more concise for reading, but that will lose some details. Hence, we just keep monthly time series for each except ATSR. On this basis, if we put everything into one plot, this plot will be too messy to read like following figure.

[Figure]

The focus of this paper is to investigate consistency of ATSR and AVHRR, while AERONET and MODIS are chosen as reference data. Hence, we make comparison of MODIS and AERONET separately to see the performance of MODIS. When it is confirmed, time series comparison will be made with AERONET or MODIS as reference separately. Please refer to fig. 4, 5, 9, and 10 in new version of manuscript. Those time series are looking much better than before.

Figures 7, 8: these are interesting, but I feel they should be expanded to make the paper more complete. They show that the monthly solar broadband-based AOD is a reasonable proxy for AOD from AERONET or MODIS. But why not also include scatter plots like figure 7 for AVHRR and AATSR? And why not include the full time series, as well as ATSR data, in figure 8? Basically the paper is about a 1983/7 and onward AOD data set over Beijing, but nowhere is a full time series of these data actually shown in the paper.

Re: As last comments, we didn't put everything into one full time series plot. Instead we made two time series, one is to validate BEM AOD and the other is to be as reference to validate AVHRR and ATSR. Please refer to fig. 6 and 8 in new version of manuscript.

Conclusions: The paper also builds on other recent work by the de Leeuw group which assessed a combination of the ATSR and MODIS sensors to get a longer-term AOD over China. This is mentioned a few times early on, but I think the conclusion should be expanded to put these studies in perspective with each other, and give some more specific thoughts/suggestions on how best to combine data records. Overall this paper has some good points but suffers from not always following through to give the bigger picture (i.e. when you have finished your investigation of the long-term climate data record, as promised in the title, what does it actually look like) ?

Re: Together with other two reviewers' comments, we added more conclusions.

---

## Author Comment (AC2) · 15 May 2019

Review of "Investigations into the Development of a Satellite-Based Aerosol Climate Data Record using ATSR-2, AATSR and AVHRR data" for Atmospheric Measurement Techniques. This paper try to discuss the feasibility of using AVHRR to continue the aerosol optical depth (AOD) record from AATSR ending in 2012 to SLSTR starting at 2016 over Beijing-Tianjing-Hebei region. The study is relevant and the potential product will benefit the aerosol community. However, there are some major issues that need to be addressed before it is suitable for publishing.

Re: Thank you for affirmation and suggestions. We have addressed these points in the revised manuscript.

1. The reason author chooses to use AVHRR is because not only can this data bridge the gap between AATSR and SLSTR but also it can extend the data record to 1983. This idea is presented in introduction, but there is only one plot Figure 8 shows the AVHRR data before 2000. All other analyses are focused 2000 to 2012. I think if we only consider this 2000-2012 period of time, there are a lot more aerosol products that can be used with much lower uncertainties. Thus, more analyses are needed to understand AVHRR through the entire data record or empirically correct AVHRR data to make it more suitable for a long term data record.

Re: As you suggest, AHVRR is not a good choice as there are some aerosol products with lower uncertainties like MODIS or MISR aerosol product. However, AVHRR is almost the only choice to look back to 1980s with expected accuracy. Correspondingly, we change the focus on making an extention (A)ATSR aerosol data record back to 1980s using AVHRR data. We will not talk about bridge of AATSR and SLTSR in this paper. Please refer to the introduction in new version of manuscript.

2. This is a paper about continue data from AATSR to SLSTR. But I didn't see a session in data talking about SLSTR.

Re: We change the primary purpose of this paper to extention (A)ATSR aerosol data record back to 1980s. Hence, we will not talk about SLSTR.

3. The author uses a very large portion of paper introducing aerosols and their facts. It is really not to the point of this article. Please make the introduction more concise.

Re: This is also mentioned by Dr. Andrew Sayer (referee 1). Please refer to the new version of manuscript in which the introduction is re-written.

4. To me it makes more sense to validate the radiance retrieved AOD against AERONET. Or maybe against MODIS to show the sampling bias. Then rely on radiance retrieved AOD to validate everything else consistently from 1983 to current.

Re: The validation of this "BEM (broadband extinction method) AOD" (the radiance retrieved AOD)

has already done by co-author Dr. Ling Sun in her paper of https://doi.org/10.1016/j.atmosenv.2015.08.042. The primary purpose of this paper has been switched to make an extention from 1987 to 1997 from ATSR. Hence, we adjust radiance AOD related time series from 1987 to 2012 as whole-time range (please refer to fig.6 and 8 in new version of manuscript). Here, we only validate AVHRR AOD with radiance retrieved AOD as the data volume for monthly ATSR AOD is too small which could lead to severe bias.

5. The title doesn't indicate the study region.
Re: The title has been revised as "Investigations into the Development of a Satellite-Based Aerosol Climate Data Record using ATSR-2, AATSR and AVHRR data over North-Eastern China from 1987 to 2012".

---

## Author Comment (AC3) · 15 May 2019

This manuscript compares aerosol retrievals from 3 different sensors (MODIS, ATSR, and AVHRR) with ground based validation data (including AERONET) over a 10x10 degree box centered on eastern China. The goal is a worthy one, the creation of a continuous climate data record of satellite-retrieved aerosol optical depth from the early 1980's until the present day period. I believe this work can be published if the authors are willing to substantially change the manuscript. Most of my comments are embedded within the PDF attached, but I will summarize a few points.
Re: Thank you for affirmation and the constructive comments. We have addressed these points in the new version of manuscript.

1) As was mentioned by the other two reviewers, where is the SLSTR data? If this data is not available, this manuscript can still add value, but not much (in its current form).
Re: The focus of this paper has been switched to extent ATSR data set back to 1980s. The content about SLSTR data is deleted in the new version of manuscript.

2) MISR shares a lot of similarities to ATSR including: swath size, multi-angle viewing, equatorial crossing time, and algorithm heritage (over-land). Additionally, I expect that the error statistics for MISR (over this region) are quite a bit better than for any other sensor used in this paper. In fact, given MISR's very long data record (2000–>2019 and counting), its similarities with ATSR, its overlap with *both* ATSRs *and* SLSTR, I think it makes much more sense to use MISR to stitch together the ATSRs and SLSTR. Once those two datasets are harmonized with MISR (globally, not for one region), I would then look back and compare with AVHRR (globally, or at least using all regions available).
Re: Thanks for this suggestion. MISR data is possibly a better choice to bridge the gap of AATSR and SLSTR data. As mentioned above, the content SLSTR data are deleted, and we have to switch the focus of this paper on period from 1987 to 2012.

3) Please find a way to make this work much more global. A 10x10 degree region is not a very useful climate data record, especially in a region with so much dust and pollution transport. Additionally, the authors could show consistencies and discrepancies with other sensors via a map of gridded correlations and differences (using seasonal AODs, compared with other sensors).
Re: Thank you for your suggestions. We start from the small regions as we only have this region AVHRR long-term AOD till now. Besides, this region with complex aerosol types helps us to analyze the effectiveness of AOD products in complex situations, because it's easy for everyone to be very accurate in simple situations. Correlation maps are not the best choice as the data volume of seasonal maps are with big differences. Hence, we plot seasonal difference maps for three products.

4) As a third party (I work with MISR data) with no stake in any of these instruments (at least data from the ones presented), it seems pretty clear from this small dataset that MODIS provides the best available AOD here (by far). One (or more) of three things is going on here: (1) AATSR's aerosol retrieval algorithm is inferior to MODIS, (2) AATSR's sample size in this region is so small as to border on the irrelevant, or (3) the region selected is so small that regional biases in the algorithm dominate your observed errors. If (1), I have to wonder why bother stitching together AOD from ATSR and AATSR with SLSTR (and AVHRR) at all? Even though ATSR, AATSR, and SLSTR all lack a blue band (which will significantly degrade performance over brighter regions), this should be compensated by the additional view angle. If the current algorithm is insufficient, maybe a new one should be developed. Otherwise, if MODIS truly gives better performance, just create the CDR using MODIS, AVHRR, and VIIRS, which would be easier anyways. If (2) or (3) see point 3) above, you need more data.

Re: The focus of this paper has been switched to make an extention of ATSR back to 1980s, and the study area is still limited to north China before newly AVHRR data are produced. The performance of ATSR is inferior to MODIS over small region but they are very close when study area is extended to whole China region as described in Sogacheva et al (2018). Besides, ATSR could provide data set before 2000. Our following work is to produce AVHRR data set over whole China.

Reply to comments in supplement:
1. Page 1, line 6, AOD>0.6 is not uncommon for the Beijing region in winter either.
Re: We revised this in the new version of manuscript.

2. Page 3, line 11, delete "and thus restore nature"
Re: We rewrote introduction in new version of manuscript in which this line was deleted.

3. Page4, line 22, Why choose only China if Europe is also available with this dataset? I would expect that Europe may have had some validation data going further back than the validation data available over China.
Re: Our following work is to produce AVHRR AOD data set over whole China region and we have BEM AOD (boardband extinction method) from at least ten sites. Hence, in this paper we focus on north China, but we will consider Europe in next paper.

4. Figure1, This figure needs to have land cover included on the colormap. We should not have to scroll up and down.
Re: Color codes have been replaced by the name of land covers.

5. Page 5, line 22, Any reason MISR is not included here? Of the sensors listed here, it probably compares most favorably with validation in the region. The latest version (23) of the product also has an improved spatial resolution (17.6-->4.4 km), and performs noticeably better at high AOD.
Re: MISR is really a good instrument for producing aerosol data set. The current purpose of this paper is to extent AOD back to 1987. When SLSTR data are available, MISR is possibly the best choice to bridge the gap of AVHRR and SLSTR. But this will be considered in another study.

6. Page 6, line 19, Sea spray extinction probably never exceeds 0.01 at 3.75 microns (and is probably

much smaller).

Re: We only consider the land surface, and sea is not included in this study of AVHRR AOD.

7. Page 7, line 6, I am curious, is this 55 degrees from the surface normal, or is boresight angle 55 degrees?

Re: 55 degrees are from nadir to forward. We have added an explanation of this in the new version of manuscript.

8. Page 7, line 9, Similar swath to MISR (400 km), similar crossing time to MISR as well.

Re: Thanks for this remind and we will use MISR data to bridge the gap between AATSR and SLSTR when SLSTR data is ready.

9. Page 7, line 11, MISR's shape-similarity algorithm (part of the land algorithm) is probably based on this heritage.

Re: Thanks again.

10. Page 11, line 6, It would be relatively simple to perform a correlation between the different sensors. Additionally, a correlation (and RMSD) map based on seasonal mean AODs could be generated as well. This could give a very good indication on spatial agreement.

Re: We have plotted seasonal AOD difference maps instead of correlation maps as the data volume in averaging seasonal AOD for ATSR is much less than the other two.

11. Page 11, line 12, delete "overall somewhat"

Re: It has been deleted.

12. Page 12, line 11, For MODIS or ATSR?

Re: It is the difference between MODIS and ATSR.

13. Page 13, line 12, Hence the importance of using data from all available regions (including Europe). This will also greatly increase the amount of data available from AATSR.

Re: The presented paper focuses on data sets over north China, but our following research will be extended to whole China regions so that the data volume of ATSR will not be a problem. We will consider Europe as well.

14. Page 13, line 13-15, replaced by "an adequate retrieval over bright surfaces".

Re: This line has been replaced by "an adequate retrieval over bright surfaces".

15. Figure 3, as a third party with no stake in any of these instruments, it seems pretty clear (from this small dataset) that MODIS provides the best available AOD here (by far). One (or more) of three things is going on here: AATSR's aerosol retrieval algorithm is inferior to MODIS, AATSR's sample size in this region is so small as to border on the irrelevant, or the region selected is so small that regional biases in the algorithm dominate your observed errors.

Re: The same as (4).

16. About qualities of figures.

Re: We will upload figures with high quality with a resolution of at 300dpi separately.

---

## Author Comment (AC4) · 15 May 2019

**Investigations into the Development of a Satellite-Based Aerosol Climate Data Record using ATSR-2, AATSR and AVHRR data over North-Eastern China from 1987 to 2012**

Yahui Che[1,7], Jie Guang[1], Gerrit de Leeuw[2,43], Yong Xue[1,34], Ling Sun[5], and Huizheng Che[6]

[1] Key Laboratory of Digital Earth Science, Institute of Remote Sensing and Digital Earth, Chinese Academy of Sciences (RADI/CAS), Beijing 100094, China
[2] Finnish Meteorological Institute, Climate Research Programme, P.O. Box 503, FI-00101 Helsinki, Finland
[3] School of Atmospheric Physics, Nanjing University of Information Science and Technology, Nanjing 210044, China
[3 4] Department of Electronics, Computing and Mathematics, College of Engineering and Technology, University of Derby, Derby DE22 1GB, UK
[4] School of Atmospheric Physics, Nanjing University of Information Science and Technology, Nanjing 210044, China
[5] Key Laboratory of Radiometric Calibration and Validation Environmental Satellites (LRCVES/CMA), National Satellite Meteorological Center, China Meteorological Administration, Beijing 100081, China
[6] State Key Laboratory of Severe Weather and Institute of Atmospheric Composition, Chinese Academy of Meteorological Sciences, CMA, Beijing 100081, China
[7] University of Chinese Academy of Sciences, Beijing, 100049, China

*Correspondence to*: Jie Guang (guangjie@radi.ac.cn) and Gerrit de Leeuw (Gerrit.Leeuw@fmi.fi)

**Abstract.** Satellites provide information on the temporal and spatial distributions of aerosols on regional and global scales. With the same method applied to a single sensor all over the world, a consistent data set is to be expected. However, the application of different retrieval algorithms to the same sensor, and the use of a series of different sensors may lead to substantial differences and no single sensor or algorithm is better than any others everywhere and at any time. For the production of long-term climate data records, the use of multiple sensors cannot be avoided. The Along Track Scanning Radiometer (ATSR-2) and the advanced ATSR (AATSR) Aerosol Optical Depth (AOD) data sets have been used to provide a global AOD data record over land and ocean of 17-years (1995-2012), which is planned to be extended with AOD retrieved from a similar sensor., i.e. the Sea and Land Surface Temperature Radiometer (SLSTR) which flies on Sentinel-3A launched in early 2016. However, this leaves a gap of about 4 years between the end of the AATSR and the start of the SLSTR data records. To fill this gap, and to 
[revised manuscript text omitted]
). ~~MODIS is approaching the end of its life time, but the AOD time series will be extended with NPP/VIIRS (cf. Levy et al., 2015). The AATSR time series will be extended with data from the operational Sea and Land Surface Temperature Radiometer (SLSTR) which was launched on Sentinel-3 in February 2016. This leaves a gap of about 4 years between AATSR and SLSTR. This gap could be filled with MODIS data, as described in Sogacheva et al. (2018b) who combined ATSR and MODIS/Terra (C6.1) data to construct an AOD time series 1995-2017to meet requirements of aerosol and other communities, this time series needs~~this

is still expected to be extended especially before the 1990s to meet requirements of other communities. The Advanced Very High Resolution Radiometer (AVHRR) onboard the NOAA series satellites series iscould be a good choice to longerextend thise ATSR time series as it started observations continuously from 1978 to the present. The Xue et al. (2017) developed an AOD data set encompassinges only two relatively small areas over Europe and China from 1983 to 2014, but covers the complete period, as opposed to the AVHRR global over land AOD data set recently presented by Hsu et al. (2017) and Sayer et al. (2017) which encompasses several distinct time periods. Hence in this study we focus here on the Xue et al (2017) AVHRR AOD data set over China and compare that with ATSR-derived AOD data to determine its suitability for merging (as done for ATSR/MODIS by Sogacheva et al. 2018b), and thus extending the ATSR data set both before the ATSR-2 era (and possibly after the AATSR was lost in 2012, although other data sets may be more suitable for to extend to later years as shown in Sogacheva et al. (2018b) for MODIS. So the focus The objective of the current study is to investigate whether thisthe ATSR AOD data set can also be doneextended to earlier years by using the Xue et al. (2017) AVHRR AOD data over China. For comparioncomparison of the ATSR and AVHRR AOD data sets, we use , using the AOD data set over land which was recently presented by Xue et al. (2017). The most common instrument used for aerosol retrieval is the Moderate Resolution Imaging Spectroradiometer (MODIS) which was launched on the Terra satellite, in a morning orbit with equator crossing time (descending) at 10:30 local time (LT), in December 1999 and on the Aqua satellite (ascending, equator crossing time 13:30 LT) in May 2002. MODIS thus provides an AOD time series since 2000 and is still operational. In this study we use both ground-based reference data, from AERONET (Holben et al., 1988) and from CARSNET (Che et al, 2009, 2015), and MODIS C6.1 AOD data. for comparison and evaluation. These Rreference data and MODIS/Terra data for inter-comparison are not available for the period before 2000 and therefore we also use AOD data derived from broadband radiation measurements using the broadband extinction methods (BEM) derived AOD data (Xu et al., 2015, Guo et al, 2016b) as described in Sect. 2.3.3. Data sets and methods used are presented in section 2. An overview of the data and an evaluation of their quality are presented in section 3, including a comparison of the various data sets. The results are discussed and conclusions are presented in section 4. era until SLSTR

Aerosol particles can be emitted by natural processes such as the interaction between wind and waves which produces sea

In addition to this large variety of sources, the aerosol concentrations vary strongly with meteorological conditions, which in turn are affected by large scale synoptic conditions and weather systems (Li et al., 2017a; Miao et al., 2017; Yim et al., 2019). The aerosol concentrations also vary with economic development such as industrialization, urbanization and the ensuing policy measures to reduce adverse effects on health and climate. Since the industrial revolution in the 18th century, the emission of air pollutants has been increasing until adverse effects were recognized and measures were developed and implemented to reduce emissions and concentrations of pollutants (e.g., Brimblecombe, 2006). In particular, in the second half of the 20th century the effects of $SO_2$ on forests and lakes, also known as acid rain, was recognized and measures were taken to reduce the $SO_2$ emissions and thus restore nature. Later, the adverse effects of $NO_2$ emissions and effects of aerosols, especially fine

particulate matter, on air quality, health and climate were recognized and reduced. These led to the reduction of air pollution in developed countries, in particular in North America and Europe (Guerreiro et al., 2014), but in developing countries with increasing industrial activity and urbanization the concentrations continued to increase (Hao et al., 2000). Examples are China and India where the concentrations of pollutants are amongst the highest in the world. Taking China as an example, recent

5  publications show the effect of policy measures on the reduction of the most polluting trace gases $SO_2$ and $NO_2$ (van der A et al., 2017), which, as precursor gases, also affect the concentrations of aerosols. In particular, the emissions of $SO_2$ were reduced as part of the 11th Five-Year Plan (2006-2010) (Zheng et al., 2018), but the emissions of $NO_2$ continued to increase (e.g., van der A et al., 2017) until the 12th Five-Year Plan (2011-2015). Large emission reductions were achieved after 2013 when the Clean Air Action was enacted and implemented and the $NO_2$ concentrations decreased (Zheng et al., 2018). Starting from 2011,

10  aerosol concentrations decreased in China as shown, e.g., from satellite observations of the aerosol optical depth (AOD) (Zhang et al., 2017, Zhao et al., 2017, de Leeuw et al, 2018, Sogacheva et al., 2018b).

Observations of the concentrations of trace gases and aerosols in China are publicly available since several observational networks have been established, such as CARE-China (Xin et al., 2015), and the NASA's AERONET (AErosol RObotic NETwork) (Holben et al. 1998) with observations mainly in the east of China, the Chinese CARSNET (Chinese Aerosol

15  Remote Sensing Network) (Che et al., 2009; 2015) and SONET (Sun-sky radiometer Observation NETwork) (Li et al., 2018), all of which provide data across the whole country. However, most of these observations were established in the last decade and very few, if any, historical data on large scale are available for the construction of the long time series needed to show the evolution of pollutant concentrations over many years and analyse the effects of different contributions. Here, satellite data may offer a solution. The most common satellites used for the observation of trace gases and aerosols offer information since

20  the beginning of the 21st century and, by combining the information from different instruments, time series encompassing two decades can be constructed (de Leeuw et al., 2018, Sogacheva et al., 2018b). Satellite information has been used together with model simulation to analyse the effects of natural and anthropogenic contributions on the concentrations of trace gases and aerosols (Kang et al., 2018). In another study combining satellite data with ground-based observations, the role of precursor gases, in particular VOCs, and photochemical reactions in the formation of aerosols ($PM_{2.5}$) was revealed (Bai et al., 2018).

25  In this study we focus on aerosols, and in particular on the AOD observed by satellites. The most common instrument used for aerosol retrieval is the Moderate Resolution Imaging Spectroradiometer (MODIS) which was launched on the Terra satellite, in a morning orbit with equator crossing time (descending) at 10:30 local time (LT), in December 1999 and on the Aqua satellite (ascending, equator crossing time 13:30 LT) in May 2002. MODIS thus provides an AOD time series since 2000 and is still operational. The Along Track Scanning Radiometer ATSR-2, a dual view instrument, was launched in 1995 on the

30  European Space Agency (ESA) satellite ERS-2 and provided data until 2003. The Advanced ATSR (AATSR) is a similar instrument launched in 2002 on the ESA platform ENVISAT, which was lost in April 2012. The AOD over China from ATSR-2 and AATSR is consistent (Sogacheva et al., 2018a) and hence, together these instruments provide a 17-year AOD time series, 1995-2012 (Popp et al., 2016, de Leeuw et al., 2018). Combining ATSR and MODIS, 22-year AOD measurements were constructed, showing the AOD increased until about 2006, and then clear decreased since 2011 over China (de Leeuw et al.,

2018, Sogacheva et al., 2018b). MODIS is approaching the end of its life time, but the AOD time series will be extended with NPP/VIIRS (cf. Levy et al., 2015). The AATSR time series will be extended with data from the operational Sea and Land Surface Temperature Radiometer (SLSTR) which was launched on Sentinel-3 in February 2016. This leaves a gap of about 4 years between AATSR and SLSTR. This gap could be filled with MODIS data, as described in Sogacheva et al. (2018b) who
5   combined ATSR and MODIS/Terra (C6.1) data to construct an AOD time series 1995-2017.  The objective of the current study is to investigate whether this can also be done by using AVHRR data, using the AOD data set over land which was recently presented by Xue et al. (2017). The advantage of using AVHRR data is the long time series which would allow extension back to the earliest AVHRR AOD retrievals available, in this case 1983, and thus construct a time series 1983-present. The Xue et al. (2017) data set encompasses only two relatively small areas over Europe and China, but covers the
10  complete period, as opposed to the AVHRR global over land AOD data set recently presented by Hsu et al. (2017) and Sayer et al. (2017) which encompasses several distinct time periods. Hence we focus here on the Xue et al (2017) AVHRR AOD data set over China and compare that with ATSR-derived AOD data to determine its suitability for merging (as done for ATSR/MODIS by Sogacheva et al. 2018b), and thus extending both before the ATSR-2 era and after the AATSR era until SLSTR. In this study we use both ground-based reference data, from AERONET (Holben et al., 1988) and from CARSNET
15  (Che et al, 2009, 2015), and MODIS C6.1 AOD data for comparison and evaluation. Reference data and MODIS/Terra data for inter-comparison are not available for the period before 2000 and therefore we also use radiance-derived AOD data (Xu et al., 2015, Guo et al, 2016b). Data sets and methods used are presented in section 2. An overview of the data and an evaluation of their quality are presented in section 3, including a comparison of the various data sets. The results are discussed and conclusions are presented in section 4.

20  **2 Method**

**2.1 Study area**

The study area is located over north-eEastern China, i.e. between 110 °and 120 °E and 35 °and 45 °N (Fig. 1), which is divided into two sub-regions by the Taihang Mountain range with the North China Plain (NCP) and large urban agglomerations like Beijing and Tianjin and the Hebei province (together BTH which is among the most populated and fast developing regions in
25  China) to the SE and the mountainous terrain to the NW extending over the Loess Plateau in Shanxi Province and the Inner Mongolia plateau. The Taihang mountain range forms a natural barrier for the transport of air pollution resulting in the frequent accumulation of pollutants and the occurrence of haze over the BTH area and the NCP (e.g., Sundström et al., 2012; Wang et al., 2013). The satellite-derived AOD maps in Fig. 2 show that this line also roughly divides high AOD in the SW of the study area and low AOD in the NW. The background in Fig. 1 is a land cover map showing that the major land cover types in the
30  study area are croplands in the SE and grassland to the NW, which are intersected by mixed forest and closed shrublands as shown in the inset in Fig. 1.

[Figure]

**Figure 1: Study area, with the locations of the ground-based reference sites discussed in Sect. 2.3 (CARSNET: red squares, AERONET: blue circles, solar radiation station: green triangle) overlaid on the IGBP land cover map.** ~~Land cover is colour coded, see the legend to the right of the map: Water (0), Evergreen Needleleaf forest (1), Evergreen Broadleaf forest (2), Deciduous Needleleaf forest (3), Deciduous Broadleaf forest (4), Mixed forest (5), Closed shrublands (6), Open shrublands (7), Woody savannas (8), Savannas (9), Grasslands (10), Permanent wetlands (11), Croplands (12), Urban and built-up (13), Cropland/Natural vegetation mosaic (14), Snow and ice (15) and Barren or sparsely vegetated (16).~~

**2.2 Satellite data**

Satellite-retrieved data sets used in this study are satellite-derived AOD data from six radiometers, i.e. AVHRR-1, AVHRR-2, AVHRR-3, ATSR-2, AATSR and MODIS. These data sets are briefly discussed below.

**2.2.1 AVHRR**

The AVHRR instruments flew on a series of satellites, most of them with an afternoon equator crossing time at 13:40 LT (ascending) (see Xue et al., 2017, for an overview). AVHRR has a swath width of 23992900 km at 833km altitude (Robel and Graumann, 2014https://earth.esa.int/web/guest/missions/3rd-party-missions/current-missions/noaa-avhrr) 
[revised manuscript text omitted]

Bond, T. C., Doherty, S. J., Fahey, D. W., Forster, P. M., Berntsen, T., DeAngelo, B. J., Flanner, M. G., Ghan, S., Kärcher, B., Koch, D., Kinne, S., Kondo, Y., Quinn, P. K., Sarofim, M. C., Schultz, M. G., Schulz, M., Venkataraman, C., Zhang, H., Zhang, S., Bellouin, N., Guttikunda, S. K., Hopke, P. K., Jacobson, M. Z., Kaiser, J. W., Klimont, Z., Lohmann, U., Schwarz, J. P., Shindell, D., Storelvmo, T., Warren, S. G., and Zender, C. S.: Bounding the role of black carbon in the climate system: A scientific assessment, J. Geophys. Res. Atmos., 118, 5380-5552, doi: 10.1002/jgrd.50171, 2013.

Boucher, O., Randall, D., Artaxo, P., Bretherton, C., Feingold, G., Forster, P., Kerminen, V.-M., Kondo, Y., Liao, H., Lohmann, U., Rasch, P., Satheesh, S.K., Sherwood, S., Stevens, B., and Zhang, X.: Clouds and Aerosols. In: Climate Change 2013: The Physical Science Basis. Contribution of Working Group I to the Fifth Assessment Report of the Intergovernmetral Panel on Climate Change, Cambridge University Press, Cambridge, United Kingdom and New York, NY, USA, 2013.

Brimblecombe, P.: The Clean Air Act after 50 years, Weather, 61, 311-314, doi: 10.1256/wea.127.06, 2006.

Carn, S. A., Fioletov, V. E., McLinden, C. A., Li, C., and Krotkov, N. A.: A decade of global volcanic SO2 emissions measured from space, Sci. Rep., 7, 44095, doi: 10.1038/srep44095, 2017.

[revised manuscript text omitted]

Zhang. L., Liu, Y. and Hao, L.: Contributions of open crop straw burning emissions to PM2.5 concentrations in China. Environ. Res. Lett., 11, 014014, doi:10.1088/1748-9326/11/1/014014, 2016.

[revised manuscript text omitted]

---

## Author Response (AR2)

Dear Editor, dear Dr. Andrew Sayer,

Thank you for giving me the opportunity to revise and resubmit this manuscript. Many thanks for the valuable comments. The detailed replies are addressed point by point and we have modified the paper in response to the reviewer comments.

Best regards,

Yahui Che on behalf of all authors

2019-6-27

**Response to Dr. Andrew Sayer:**

1. Section 2.2.1 – the text reads like the Xue and Deep Blue AVHRR data sets are the only land ones (e.g. page 5 lines 17). But there's also some (in various states of availability) by groups including Mei et al, Gao et al, Hauser et al, etc.

Response: We have revised this sentence by adding more citations in the revised version of manuscript. The related statement in the revised draft is as follow: "Several AOD retrieval algorithms applied to AVHRR observations over land were published, including Hauser et al., (2005), Li et al., (2011), Mei et al., (2014), Xue et al. (2017), Sayer et al. (2017), Hsu et al (2017), and Gao et al., (2018)."

2. Page 9 line 24: I would not described 0.06 as a "small" bias, and suggest removing that word.

Response: We have deleted "small" in this line in the revised version of manuscript.

3. Figure 2 (and later): Should the middle column say "ATSR_ADV" instead of "AVHRR_ADV"? It would also be better to have captions without underscores, e.g. write "AVHRR ADL AOD, JJA" instead of "AVHRR_ADL_AOD_JJA". The algorithm abbreviations could also be dropped, since only one algorithm is applied to each sensor here.

Response: Thank you for pointing out this typo. We have corrected all errors in subtitles and adjusted them as you suggested in Fig. 2 and 9.

4. Figure 3: The caption refers to least squares fit parameters. This text should be removed because these are not shown in the figure (to be clear: they should not be added to the figure; the caption should just be shortened).

Response: We put LSQ equation onto each scatter panel in the first version of manuscript, but later we thought EE should be emphasized for each scatter. So we removed relevant parameters but didn't remove least squares words. "LSQ fit" has been deleted in the current revision of manuscript. We have shortened the caption by moving part into the text.

5. Table 2: likewise "for a LSQ fit to all data points" can be deleted.

Response: We have deleted "for a LSQ fit to all data points" in the new version of the manuscript.